# Synthetically-primed adaptation of *Pseudomonas putida* to a non-native substrate D-xylose

Pavel Dvořák [1,9] ✉, Barbora Burýšková [1,9], Barbora Popelářová [1,9], Birgitta E. Ebert [2,9], Tibor Botka [1], Dalimil Bujdoš [3,4], Alberto Sánchez-Pascuala [5], Hannah Schöttler [6], Heiko Hayen [6], Víctor de Lorenzo [7], Lars M. Blank [8] & Martin Benešík[1]

To broaden the substrate scope of microbial cell factories towards renewable substrates, rational genetic interventions are often combined with adaptive laboratory evolution (ALE). However, comprehensive studies enabling a holistic understanding of adaptation processes primed by rational metabolic engineering remain scarce. The industrial workhorse *Pseudomonas putida* was engineered to utilize the non-native sugar D-xylose, but its assimilation into the bacterial biochemical network via the exogenous xylose isomerase pathway remained unresolved. Here, we elucidate the xylose metabolism and establish a foundation for further engineering followed by ALE. First, native glycolysis is derepressed by deleting the local transcriptional regulator gene *hexR*. We then enhance the pentose phosphate pathway by implanting exogenous transketolase and transaldolase into two lag-shortened strains and allow ALE to finetune the rewired metabolism. Subsequent multilevel analysis and reverse engineering provide detailed insights into the parallel paths of bacterial adaptation to the non-native carbon source, highlighting the enhanced expression of transaldolase and xylose isomerase along with derepressed glycolysis as key events during the process.

Efficient utilization of the most abundant lignocellulose-derived sugars (D-glucose, D-xylose, L-arabinose) and aromatic chemicals (e.g., *p*-coumarate, ferulate) is a key prerequisite for the economic viability of biotechnological processes that leverage natural or recombinant microorganisms as biocatalysts[1–4]. D-Xylose is a predominant monomeric hemicellulose component, which forms 10–50% of the mass of lignocellulosic residues[5]. However, many biotechnologically relevant microorganisms lack xylose metabolism (*Zymomonas mobilis*, *Corynebacterium glutamicum*, *Saccharomyces cerevisiae*), while others can utilize it but prefer glucose or other available organic substrates as their primary carbon and energy source (e.g., *Escherichia coli*, *Bacillus subtilis*)[6]. The desirable phenotype can be established using metabolic

[1]Department of Experimental Biology, Faculty of Science, Masaryk University, Kamenice 753/5, 62500 Brno, Czech Republic. [2]Australian Institute for Bioengineering and Nanotechnology, The University of Queensland, Cnr College Rd & Cooper Rd, St Lucia, QLD QLD 4072, Australia. [3]APC Microbiome Ireland, University College Cork, College Rd, Cork T12 YT20, Ireland. [4]School of Microbiology, University College Cork, College Rd, Cork T12 Y337, Ireland. [5]Department of Biochemistry and Synthetic Metabolism, Max Planck Institute for Terrestrial Microbiology, Karl-von-Frisch-Straße 10, 35043 Marburg, Germany. [6]Institute of Inorganic and Analytical Chemistry, University of Münster, Corrensstraße 48, 48149 Münster, Germany. [7]Systems and Synthetic Biology Program, Centro Nacional de Biotecnología CNB-CSIC, Cantoblanco, Darwin 3, 28049 Madrid, Spain. [8]Institute of Applied Microbiology, RWTH Aachen University, Worringer Weg 1, 52074 Aachen, Germany. [9]These authors contributed equally: Pavel Dvořák, Barbora Burýšková, Barbora Popelářová, Birgitta E. Ebert. ✉e-mail: pdvorak@sci.muni.cz

engineering tools[7]. The rational approach to metabolic engineering employs our vast yet incomplete knowledge of the cellular machinery to introduce non-native functions into microbial cell factories. However, the genetic instalment of new functions can cause imbalances in the metabolic network and is therefore often completed with natural selection[8–11]. By harnessing the natural selection of the fittest, adaptive laboratory evolution (ALE) on the substrate(s) of choice can generate mutants with optimized functioning of the rewired metabolism.

Gram-negative bacteria from the genus *Pseudomonas* are sought after in biotechnology, particularly for their robustness, fast growth in inexpensive media, and versatile metabolism[12–14]. *P. putida* KT2440 is one of the most studied pseudomonads, especially in the biodegradation and bioremediation field. Its capacity to degrade lignin and stream the resulting aromatic monomers into biomass and diverse bioproducts makes it a promising candidate for lignocellulose valorization[15–17]. Still, a major drawback is *P. putida* KT2440´s inability to utilize pentose sugars such as D-xylose or L-arabinose.

To overcome this bottleneck, *P. putida* has previously been endowed with different exogenous xylose utilization pathways[18–23]. The isomerase route is the most promising for biotechnological xylose valorization[4,22]. It is composed of xylose isomerase XylA and xylulokinase XylB and its product xylulose 5-phosphate (X5P) enters directly into the EDEMP cycle, a pathway configuration specific to *P. putida* (Fig. 1)[24]. In our former study, we found that *P. putida* EM42, the

genome-reduced derivative of the strain KT2440[25], requires the XylE xylose/H[+] symporter from *E. coli* and the deletion of the *gcd* gene encoding periplasmic membrane-bound glucose dehydrogenase (PP_1444) to fully utilize xylose[18]. This is consistent with the findings of Meijnen et al.[20,26]. in *P. putida* S12 and Elmore and co-workers[19] in *P. putida* KT2440. It was also demonstrated that recombinant *P. putida* can fully co-utilize xylose with glucose and other lignocellulosic sugars and aromatics[4,18,19,27,28]. *P. putida*, empowered with an efficient xylose metabolism based on the exogenous isomerase route represents a very attractive biocatalyst for the production of polyhydroxyalkanoates, *cis,cis*-muconate and other dicarboxylic acids, biosurfactants, some amino acids (e.g., threonine), and potentially many other compounds that can be synthesized from xylose alone or from complex lignocellulosic hydrolysates[4,22,29]. However, the growth of our recombinant *P. putida* on xylose was still slow[18]. Moreover, the xylose assimilation into the bacterium´s biochemical network extended with the isomerase route was unresolved. This complicated further tailoring of the utilization and valorization of this sugar by *P. putida*.

In the present study, we use an evolutionary approach primed by rational metabolic engineering and introduction of synthetic genetic constructs to (i) explore how the metabolism of *P. putida* adapts to the perturbations caused by carbon flux from a non-native sugar substrate, and (ii) accelerate xylose utilization in our previously reported *P. putida* EM42 PD310 strain (Fig. 1, Table 1). Carbon flux analyses and enzyme activity assays help us uncover xylose metabolism in *P. putida*, remove the repression of part of the EDEMP cycle, and identify potentially problematic nodes in the pentose phosphate pathway (PPP). ALE of strains with or without 6-phosphogluconate dehydrogenase and with implanted exogenous PPP genes doubles the growth rate and reduces the lag phase up to 5-fold. Genomic, proteomic, and enzyme activity analyses of mutant strains together with reverse engineering reveal that the bacterium´s genomic and metabolic plasticity enabled it to find alternative solutions that led to equally improved phenotypes of selected mutants. This study elucidates xylose metabolism in *P. putida* and contributes to a better understanding of the adaptation of bacterial metabolism to a non-native substrate.

## Results and discussion
### Elucidation of xylose metabolism in recombinant *P. putida*
Our previously constructed strain *P. putida* PD310[18] (Fig. 1) grew four-fold slower on xylose (μ = 0.11 h[−1]) than on glucose and exhibited a 5-times longer lag phase on the pentose sugar (~10 h) (Table 2, Supplementary Fig. 1a, b). Studies from other groups demonstrated faster growth (μ ~0.3 h[−1]) of engineered and evolved *P. putida* gcd[−] xylABE[+] on xylose[19,20], suggesting that *P. putida* has the capacity to utilize xylose more efficiently than observed for PD310. The authors proposed complex metabolic and regulatory re-arrangements in an evolved recombinant strain of *P. putida* S12 (including altered glucose metabolism regulation) and emphasized the role of the upregulated XylE transporter in promoting the growth of evolved *P. putida* KT2440[19,20]. We expected sufficient expression of the *xylE* gene in PD310, as the gene is controlled by the strong constitutive P_EM7 promoter and a strong synthetic ribosome binding site (RBS)[30]. Expression of heterologous enzymes and transporters can place a burden on bacterial metabolism[31]. However we hypothesized that this was not the major limitation of our strain's capacity to utilize xylose because it grew well on glucose (μ = 0.46 h[−1]; Table 2).

Therefore, we moved our attention to potential bottlenecks in the central carbon metabolism of *P. putida* and mapped the catabolism of xylose in PD310 using [13]C-based metabolic flux analysis (MFA)[32]. Distribution of carbon fluxes was previously determined for *P. putida* KT2440 or its derivatives cultured on native substrates glucose, benzoate, glycerol, gluconate, or succinate[24,33–36] but never for xylose-assimilation via a heterologous isomerase pathway. The MFA performed with 1,2-[13]C

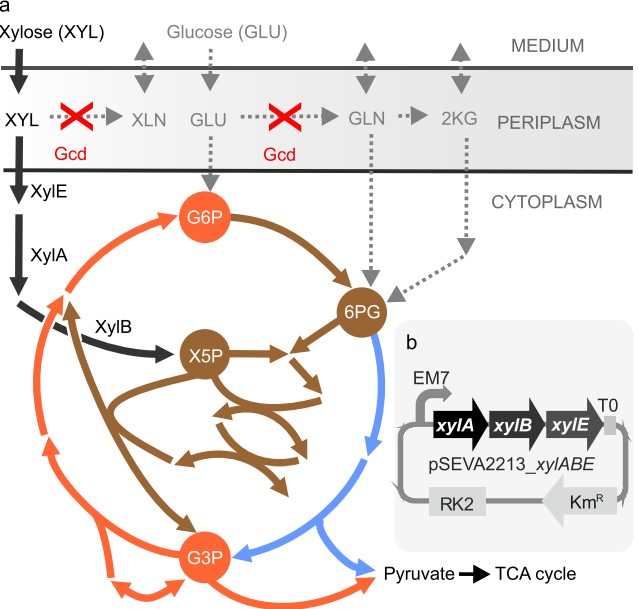

**Fig. 1 | *Pseudomonas putida* EM42 PD310 used as a template strain in this study.** Schematic illustration of (**a**) upper sugar metabolism of *P. putida* EM42 PD310 and (**b**) pSEVA2213_*xylABE* plasmid construct. The PD310 strain[18] capable of growth on D-xylose and its co-utilization with D-glucose bears low-copy-number plasmid pSEVA2213 with constitutive P_EM7 promoter and a synthetic operon that encodes exogenous xylose isomerase pathway (XylA xylose isomerase and XylB xylulokinase) and xylose/H[+] symporter XylE from *Escherichia coli* BL21(DE3). The *xylA* and *xylE* genes are preceded by synthetic ribosome binding sites (RBS), while *xylB* was left with its native RBS. Note that the elements in the plasmid scheme are not drawn to scale. The PD310 strain was also deprived of the *gcd* gene, which encodes periplasmic glucose dehydrogenase (PP_1444) to prevent the transformation of xylose to the dead-end product xylonate (XLN). The exogenous pathway converts xylose to xylulose 5-phosphate (X5P), which enters the EDEMP cycle formed by the reactions of the pentose phosphate pathway (brown arrows), the Embden-Meyerhof-Parnas pathway (orange arrows), and the Entner-Doudoroff pathway (blue arrows). Abbreviations: GLN, gluconate; G6P, glucose 6-phosphate; G3P, glyceraldehyde 3-phosphate; Km, kanamycin; 2KG, 2-ketogluconate; 6PG, 6-phosphogluconate; RK2, a broad-host-range origin of replication; T0, transcriptional terminator.

**Table 1 | *Pseudomonas putida* strains used in this study**

| Strain | Characteristics | Reference |
|---|---|---|
| EM42 | Genome-reduced derivative of strain *P. putida* KT2440: Δprophages1, 2, 3, 4 ΔTn7 Δ*endA1* Δ*endA2* Δ*hsdRMS* Δfla-gellum ΔTn*4652* | 25 |
| EM42 Δ*gcd* | Strain EM42 with deletion of *gcd* gene (PP_1444) encoding periplasmic glucose dehydrogenase | 18 |
| EM42 Δ*gcd* pSEVA2213_*xylABE* | EM42 Δ*gcd* freshly transformed with pSEVA2213_*xylABE* plasmid bearing synthetic *xylABE* operon encoding XylA xylose isomerase, XylB xylulokinase, XylE xylose-H⁺ symporter from *E. coli* (*Eco*RI/*Hind*III), see plasmid characteristics in Table S2 in Supplementary Information | This study |
| PD310 | EM42 Δ*gcd* with plasmid pSEVA2213_*xylABE*, ~118 kbp multiplication (PP_2114 - PP_2219) in chromosome | 18 |
| PD505 | EM42 Δ*gcd* with additional deletion of gene *edd* (PP_1010) encoding phosphogluconate dehydratase, with pSE-VA2213_*xylABE* plasmid | This study |
| PD506 | EM42 Δ*gcd* with additional deletion of gene *gnd* (PP_4043) encoding 6-phosphogluconate dehydrogenase, with pSEVA2213_*xylABE* plasmid | This study |
| PD507 | EM42 Δ*gcd* with additional deletions of genes *pgi*-I (PP_1808) and *pgi*-II (PP_4701) encoding glucose 6-phosphate isomerase, with pSEVA2213_*xylABE* plasmid | This study |
| PD580 | EM42 Δ*gcd* with additional deletion of gene *hexR* (PP_1021) encoding DNA-binding transcriptional regulator | This study |
| PD580 *rpoD*\* | PD580 with Ser552Pro mutation in the *rpoD* gene encoding the RNA polymerase sigma factor σ⁷⁰ | This study |
| PD584 | EM42 Δ*gcd* Δ*hexR* with pSEVA2213_*xylABE*, ~118 kbp multiplication in chromosome | This study |
| PD584 L3 | Derivative of PD584 obtained after adaptive laboratory evolution (ALE) on xylose, ~118 kbp multiplication in chromosome | This study |
| PD584 tt L3 | Derivative of PD584 obtained after ALE on xylose, ~118 kbp multiplication in chromosome | This study |
| PD689 | EM42 Δ*gcd* Δ*hexR* Δ*gnd* with pSEVA2213_*xylABE* | This study |
| PD689 *rpoD*\* | PD689 with Ser552Pro mutation in the *rpoD* gene | This study |
| PD689 tt L1 | Derivative of PD689 with chromosomally integrated expression cassette (P_EM7-*talB-tktA*-Sm^R bearing genes for trans-aldolase and transketolase from *E. coli*) obtained after ALE on xylose | This study |
| PD855 | Reverse engineered strain: PD580 with pSEVA438_*tal* and mutated pSEVA2213_*xylABE* plasmid isolated from PD584 L3 | This study |

P_EM7 constitutive promoter EM7, *Sm*, streptomycin.

**Table 2 | Growth parameters of *Pseudomonas putida* EM42 mutant strains in batch cultures with D-glucose or D-xylose as a sole carbon source**

| Strain and carbon source | $\mu$ max (h⁻¹)¹ | $Y_{X/S}$ (g_CDW g_S⁻¹)² | $q_S$ (mmol_S g_CDW⁻¹ h⁻¹)² | Lag phase (h)¹ |
|---|---|---|---|---|
| EM42Δ*gcd* xylABE⁺ XYL | 0.09 ± 0.00 | n.d. | n.d. | 41.94 ± 0.72 |
| PD310 GLU | 0.46 ± 0.01 | 0.38 ± 0.02 | 6.33 ± 0.42 | 1.40 ± 0.14 |
| PD310 XYL | 0.11 ± 0.00 | 0.31 ± 0.03 | 1.98 ± 0.32 | 10.04 ± 0.53 |
| PD506 XYL | 0.07 ± 0.00 | n.d. | n.d. | 31.72 ± 0.64 |
| PD584 XYL | 0.13 ± 0.00 | 0.35 ± 0.01 | 2.22 ± 0.16 | 4.60 ± 0.44 |
| PD689 XYL | 0.10 ± 0.00 | 0.31 ± 0.02 | 1.14 ± 0.06 | 18.25 ± 1.28 |
| PD584 L3 XYL | 0.21 ± 0.01 | 0.41 ± 0.07 | 3.07 ± 0.46 | 3.40 ± 0.60 |
| PD689 tt L1 XYL | 0.21 ± 0.00 | 0.37 ± 0.02 | 3.86 ± 0.10 | 3.70 ± 0.35 |
| PD855 | 0.20 ± 0.00 | 0.35 ± 0.03 | 3.46 ± 0.37 | 4.94 ± 0.39 |

GLU D-glucose, XYL D-xylose, *n.d.* not determined.

¹The maximal specific growth rate $\mu$ max and growth lag were calculated from growth data obtained from cultures in 48-well plates using the deODorizer program⁸⁰. Presented values are means from six biological replicates ± standard deviation.

²The biomass yield ($Y_{X/S}$) and specific carbon uptake rate ($q_S$) were calculated from data obtained from cultures in Erlenmeyer flasks. Shown values are means from three biological replicates ± standard deviation.

Source data are provided as a Source Data file.

D-xylose identified a partially cyclic upper xylose metabolism in PD310 (Fig. 2a). The majority (89%) of the carbon that initially entered the non-oxidative branch of the pentose phosphatepath-way (PPP) via xylulose 5-phosphate (X5P) was converted into fructose 6-phosphate (F6P) and further to 6-phosphogluconate (6PG) through the reactions of glucose 6-phosphate isomerase (Pgi-I and Pgi-II), glucose 6-phosphate 1-dehydrogenase (ZwfA, ZwfB, ZwfC), and 6-phosphogluconolactonase (Pgl).

This is in contrast to the flux distribution in wild-type *P. putida* KT2440 grown on glucose, which is preferentially utilized via the periplasmic oxidative route and directly funneled into the Entner-Doudoroff (ED) pathway²⁴,³³,³⁵. At the 6PG node, the flux from xylose branches. Over 50% of the carbon enters the ED pathway while more than one-third is cycled back into the non-oxidative PPP via the 6-phosphogluconate dehydrogenase (Gnd) reaction to replenish the ribulose 5-phosphate (Ru5P) and ribose 5-phosphate (R5P) pools (Fig. 2a). This partial carbon cycling via Gnd was previously suggested also by Meijnen et al., for engineered, xylose-adapted *P. putida* S12 based on transcriptome data²⁶. Here, we hypothesize that the cycling compensates for the relatively weak flux through the ribulose-5-phosphate 3-epimerase (Rpe) reaction. The bifurcation observed in our study reduced the ED pathway flux in xylose-grown PD310 (Fig. 2a) compared to wild-type *P. putida* cultured on glucose²⁴,³³,³⁵.

In line with Meijnen et al.²⁶, we determined an operational glyoxylate shunt (isocitrate lyase AceA and malate synthase GlcB) in PD310 (Fig. 2a). Glyoxylate shunt activity was reported in *P. putida* KT2440 grown on glycerol³⁴, benzoate³⁷, or a mixture of glucose and succinate³⁸ but the shunt is typically inactive during growth on glucose as the sole carbon source²⁴,³³,³⁵,³⁹. Meijnen et al. attributed glyoxylate shunt activation to the level of reducing cofactors, which was increased compared to cells cultured on glucose²⁶. Given the high fluxes measured for the four central dehydrogenases – Zwf, Gnd, pyruvate dehydrogenase complex Pdh, and malate dehydrogenase Mdh (the middle two enzymes also have decarboxylating activity) – we hypothesize that the same holds true for strain PD310 (Fig. 2a). Over-production of reducing cofactors in these reactions would also explain the negligible flux through the part of EMP pathway with fructose 1,6-bisphosphatase (Fbp), fructose 1,6-bisphosphate aldolase (Fba), and triose phosphate isomerase (TpiA) (Fig. 2a). On glucose or glycerol, gluconeogenic operation of this section of the EDEMP cycle partially recycles triose phosphates back into hexose phosphates and secures the resistance of *P. putida* to oxidative stress through a supply of

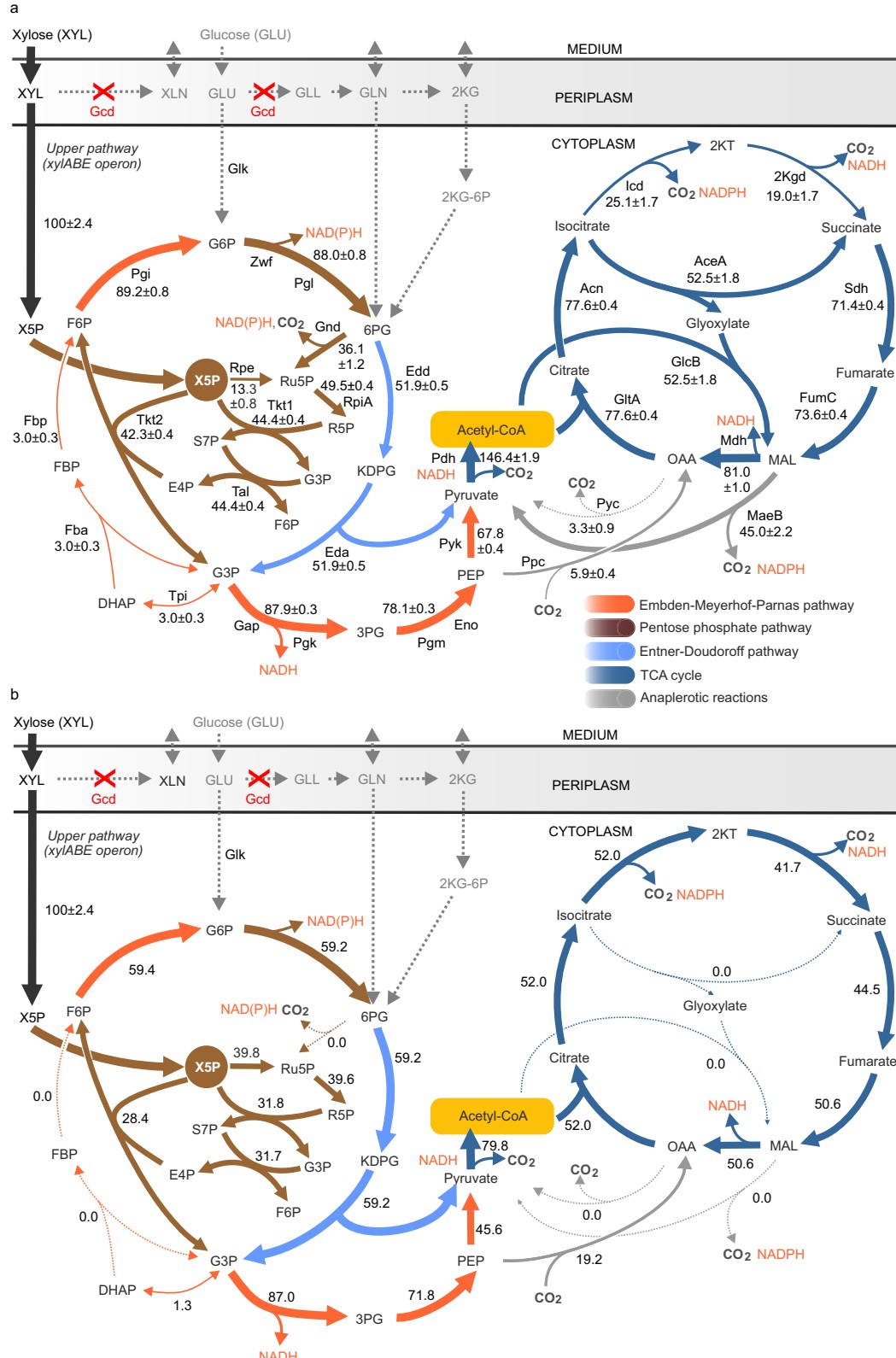

NADPH (generated by Zwf)[24,33–35,40]. However, such gluconeogenic operation of Fbp, Fba, and TpiA would be redundant to PPP activity in xylose-grown PD310.

MFA confirmed that malic enzyme MaeB activity significantly contributed to the pyruvate pool (Fig. 2a) similar to previous analyses on glucose[24,33,35]. However, xylose metabolism differs in its low pyruvate carboxylase Pyc activity.

To compute metabolic fluxes supporting optimal growth on xylose, we conducted flux balance analysis (FBA)[41] using growth rate as objective function (Fig. 2b). Growth-optimised fluxes predicted by FBA showed several differences to the MFA data, including an almost evenly distributed flux from X5P into reactions of Rpe, Tkt, and Tal, and zero flux through Gnd, the glyoxylate shunt (AceA, GlcB), and the malic enzyme MaeB reaction in FBA. Enhanced fluxes from X5P to Ru5P

**Fig. 2 | Distribution of carbon fluxes in engineered strain *Pseudomonas putida* EM42 PD310 grown on xylose.** Distribution of carbon fluxes in *P. putida* EM42 PD310 grown on xylose as determined by [13]C metabolic flux analysis (MFA, **a**) and flux balance analysis (**b**). Fluxes are given as a molar percentage of the mean specific xylose uptake rate $q_S = 1.45$ mmol $g_{CDW}^{-1}$ $h^{-1}$, which was set to 100%. Arrow thickness roughly corresponds with the given flux value. Flux values calculated in MFA (**a**) represent the mean ± standard deviation from two biological replicates ($n = 2$). Abbreviations (enzymes): AceA isocitrate lyase, Acn aconitate hydratase, Eda 2-keto-3-deoxy-6-phosphogluconate aldolase, Edd, 6-phosphogluconate dehydratase, Eno phosphopyruvate hydratase, Fba fructose-1,6-bisphosphate aldolase, Fbp fructose-1,6-bisphosphatase, Gap glyceraldehyde-3-phosphate dehydrogenase, Gcd glucose dehydrogenase, Gnd 6-phosphogluconate dehydrogenase, GlcB malate synthase, GltA citrate synthase, Icd isocitrate dehydrogenase, 2Kgd 2-ketoglutarate dehydrogenase, MaeB malic enzyme, Mdh malate dehydrogenase, Pdh pyruvate dehydrogenase, Pgi glucose-6-phosphate isomerase, Pgk phosphoglycerate kinase, Pgl 6-phosphogluconolactonase, Pgm phosphoglycerate mutase, Ppc phosphoenolpyruvate carboxylase, Pyc pyruvate carboxylase, Pyk pyruvate kinase, Rpe ribulose-5-phosphate 3-epimerase, RpiA ribose-5-phosphate isomerase, Sdh succinate dehydrogenase, Tal transaldolase, Tkt transketolase, Tpi triosephosphate isomerase, Zwf glucose-6-phosphate dehydrogenase. Abbreviations (metabolites): DHPA dihydroxyacetone phosphate, E4P erythrose 4-phosphate, FBP fructose 1,6-bisphosphate, F6P fructose 6-phosphate, GLL glucono-δ-lactone, GLN gluconate, G3P glyceraldehyde 3-phosphate, G6P glucose 6-phosphate, KDPG 2-keto-3-deoxy-6-phosphogluconate, 2KG 2-ketogluconate, 2KG-6P 2-ketogluconate 6-phosphate, 2KT α-ketoglutarate, MAL malate, NADH reduced nicotinamide adenine dinucleotide, NADPH reduced nicotinamide adenine dinucleotide phosphate, OAA oxaloacetate, PEP phosphoenolpyruvate, 3PG 3-phosphoglycerate, 6PG 6-phosphogluconate, R5P ribose 5-phosphate, Ru5P ribulose 5-phosphate, S7P sedoheptulose 7-phosphate, XLN xylonate, X5P xylulose 5-phosphate. Note that F6P and G3P are products of both Tal and Tkt reactions. Source data are provided as a Source Data file.

and higher TCA cycle activity in the FBA simulation probably fully replaced the carbon cycling via Gnd observed in vivo. FBA also computed a reduced formation of hexoses F6P and G6P (by ~33%) and, consequently, a lower flux through glucose 6-phosphate dehydrogenase Zwf.

Importantly, FBA computed approximately 41% higher growth rate ($\mu = 0.12$ $h^{-1}$) when constrained with the specific xylose uptake rate determined during the [13]C labeling experiment ($q_S = 1.45$ mmol $g_{CDW}^{-1}$ $h^{-1}$) than experimentally observed with PD310 cells ($\mu = 0.08$ $h^{-1}$ ± 0.02; note that the growth rate and xylose uptake rate in the MFA experiment differ from the values reported in Table 2 due to different experimental conditions, while the biomass yield on carbon was the same in both setups). In addition, the comparison of the sum of fluxes through oxidoreductase reactions that generate the reducing equivalents NADPH and NADH revealed 1.6-fold (NADPH) and 1.5-fold (NADH) higher values in the MFA model (Supplementary Data 1). The net rate of $CO_2$ production determined in MFA was also higher (1.6-fold) than the net rate calculated by FBA (Supplementary Data 1). As mentioned above, the surplus of NAD(P)H and loss of carbon in the form of $CO_2$ is caused by high fluxes through Zwf and Gnd reactions in PD310. The high flux through decarboxylating enzymes such as Gnd also contributes to the lower carbon efficiency of PD310 strain grown on xylose compared to glucose (see $Y_{X/S}$ biomass yields in Table 2)[42]. FBA with the genome-scale metabolic model has some limitations; it cannot model enzyme capacity, burden of protein synthesis, and some other cellular constraints. Nevertheless, a comparison of its results to the MFA showed that the flux distribution of PD310 grown on xylose was wasteful.

Next, we verified the importance of EDEMP cycle enzymes that were identified by the flux analyses as key for the growth of PD310 on xylose. Genes encoding Pgi-I (PP_1808), Pgi-II (PP_4701), Gnd (PP_4043), and Edd (PP_1010) were knocked out in *P. putida* EM42 Δ*gcd* and the growth of the resulting mutants with inserted pSEVA2213_*xylABE* plasmid (named PD505, PD506, PD507, Table 1) was tested on solid and in liquid medium (Fig. 3). These experiments demonstrated that none of the deletions was detrimental for growth on citrate, representing a gluconeogenic growth regime (Fig. 3b) and only the *edd* deletion disabled growth on glucose. In contrast, growth on xylose was affected by all three deletions. The Δ*gcd* Δ*pgi*-I Δ*pgi*-II (PD507) and Δ*gcd* Δ*edd* (PD505) mutants did not show any growth in a liquid medium within three days, confirming the essentiality of Pgi and Edd for xylose catabolism. Interestingly, the Δ*gcd* Δ*gnd* mutant (PD506) grew on xylose but with a reduced growth rate and substantially prolonged lag phase when compared to PD310 (Table 2, Fig. 3c). This result showed that the carbon flux into the non-oxidative branch of PPP through Gnd is not essential for xylose utilization by *P. putida*, in line with the FBA data.

To deepen our insight into the operation of the EDEMP cycle in PD310 grown on xylose, we screened the activities of eight enzymes, which contribute to the cycle together with the activities of the exogenous XylA and XylB (Fig. 3d, Supplementary Method 1). The specific activity of both XylA and XylB was significantly higher ($p < 0.05$) on xylose than on glucose substrate. This difference may reflect growth condition dependent epigenetic regulation of the respective genes[43]. Activities of XylB, Gnd, Zwf, Pgi, and Edd-Eda in cells cultivated on either sugar were higher than the activities in cells grown in LB medium. In the case of Zwf and Edd-Eda, the difference can be attributed to the (de)repression of these genes in cells consuming glucose or xylose. Their genes are placed in operons controlled by the HexR transcriptional repressor, which binds 2-keto-3-deoxy-6-phosphogluconate (KDPG), an intermediate of the ED pathway formed during catabolism of glucose and xylose (Fig. 2)[44]. This interaction causes HexR dissociation, which leads to transcriptional activation of these operons. The de-repression seemed to be more efficient on glucose than on the non-native substrate xylose (Fig. 3d). The low activities of Zwf and the ED pathway enzymes in cells grown on xylose (compared to glucose) could pose a bottleneck for pentose catabolism in this part of the EDEMP cycle. In contrast, the activities of transketolase, transaldolase, and Tpi were comparable across the three tested conditions, which indicates that the levels of these enzymes in *P. putida* cells are relatively constant irrespective of the substrate.

The metabolic flux analyses, gene deletion experiments, and enzyme activity measurements elucidated carbon distribution in the EDEMP cycle of xylose-grown PD310. Interestingly, the distribution of fluxes determined by MFA – especially in the PPP segment of the EDEMP cycle – resembled the situation in glucose-grown *P. putida* under oxidative stress[40]. Given that the complex biochemical network of *P. putida* is evolved primarily towards the utilization of organic acids, aromatic compounds, or glucose[45], it is plausible that the introduction of an exogenous xylose isomerase pathway led to metabolic imbalances and, consequently, to slow growth differing significantly from rates attainable on native substrates.

## Synthetically-primed enhancement of xylose metabolism in *P. putida*

Del Castillo et al.[46] and Bentley et al.[47] previously showed that the deletion of *hexR* gene de-represses *zwf-pgl-eda* and *edd-glk* operons and improves growth on glucose (Fig. 4a). Down-regulation of *hexR* was identified also in the evolved *P. putida* S12 and linked to its boosted xylose-utilization phenotype[26]. Therefore, we argued that the removal of the repressor could have a positive effect on xylose metabolism in PD310. The *hexR* gene (PP_1021) was deleted in *P. putida* EM42 Δ*gcd* and the resulting strain *P. putida* EM42 Δ*gcd* Δ*hexR* pSEVA2213_*xylABE*, named here PD584, showed improved growth compared to PD310 (Table 2, Fig. 4d, Supplementary Fig. 1c). Notably, the lag phase on

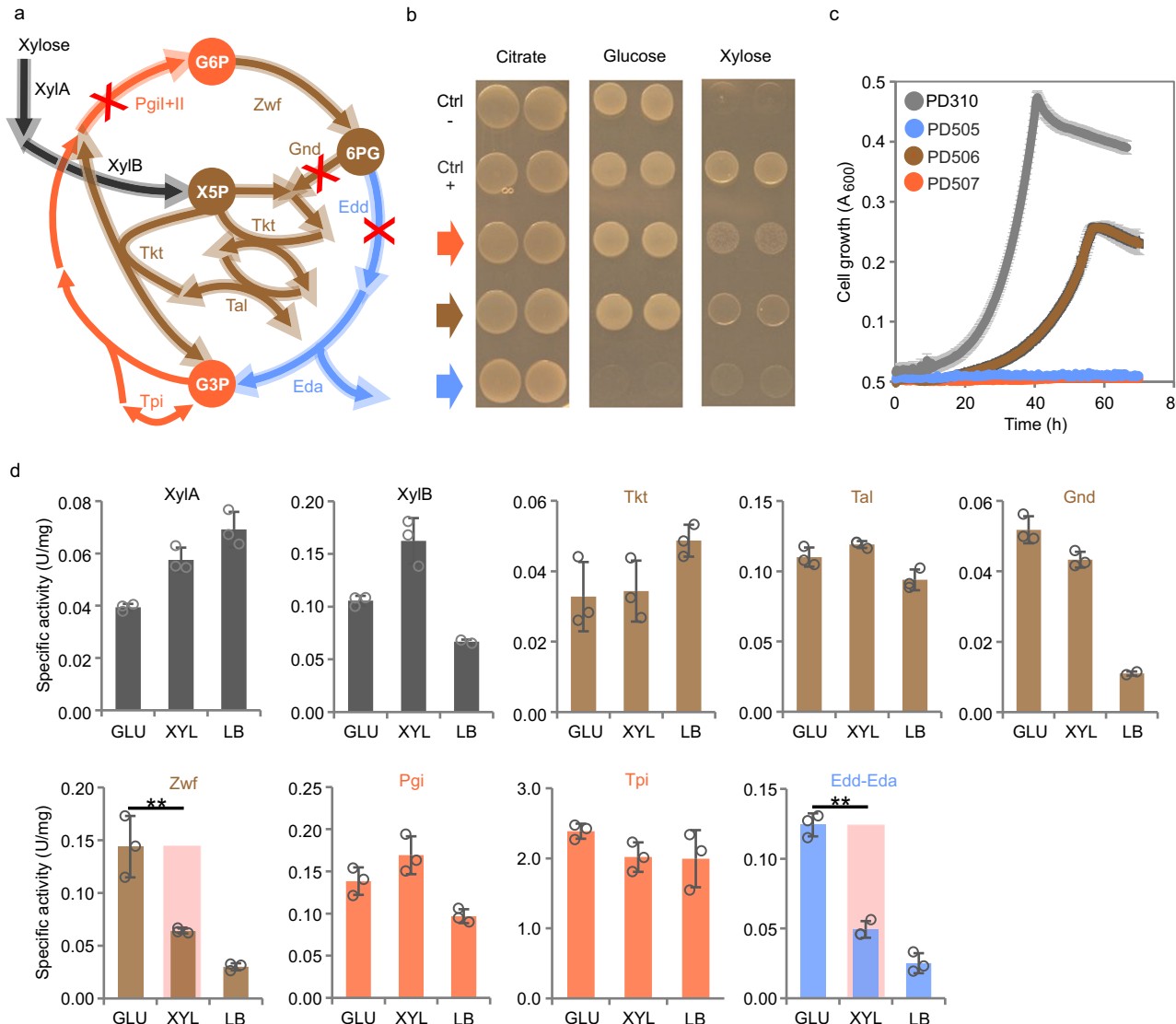

**Fig. 3 | Growth of three *Pseudomonas putida* deletion mutants on selected carbon sources and activities of XylA, XylB, and selected EDEMP cycle enzymes in *P. putida* PD310 cells. a** Scheme of the EDEMP cycle with the incorporated xylose isomerase pathway (abbreviations and color coding are the same as in Figs. 1, 2) and highlighted reactions that were eliminated by knocking out the respective genes (red crosses). **b** Growth of PD505 (Δ*gcd* Δ*edd*, green arrow), PD506 (Δ*gcd* Δ*gnd*, brown arrow), and PD507 (Δ*gcd* Δ*pgi*-I Δ*pgi*-II, red arrow) mutants on solid M9 agar medium with 2 g L$^{-1}$ citrate, D-glucose, or D-xylose used as the sole carbon and energy source. Ctrl- stands for negative control (*P. putida* EM42 Δ*gcd* with empty pSEVA2213) and Ctrl+ stands for positive control (PD310). Cells pre-cultured in LB medium were washed with M9 medium and 10 μL of cell suspension of OD 2.0 was dropped on the agar and incubated for 24 h (agar with glucose or citrate) or 48 h (agar with xylose) at 30 °C. **c** Growth of the three mutants and control PD310 in M9 minimal medium with 2 g L$^{-1}$ D-xylose in a 48-well microplate. Data points are shown as mean ± standard deviation from three (*n* = 3) biological replicates. **d** Specific activities of 10 selected enzymes (activity of Eda and Edd was measured in a combined assay) were determined in cell-free extracts (CFE) prepared from PD310 cells cultured till mid-exponential phase in rich LB medium (LB), in M9 medium with 2 g L$^{-1}$ glucose (GLU), or in M9 medium with 2 g L$^{-1}$ xylose (XYL) as detailed in Supplementary Method 1 in Supplementary Information. Data are shown as mean ± standard deviation from three (*n* = 3) biological replicates. Asterisks (**) denote statistically significant difference between two means at *p* < 0.01 calculated using two-tailed Student *t* test (*p* values = 4.07 × 10$^{-2}$ for Zwf and 3.73 × 10$^{-4}$ for Edd-Eda). Source data are provided as a Source Data file.

xylose was reduced by more than 2-fold, from 10.0 to 4.6 h. The significantly increased activities of Zwf and the ED pathway in PD584 on xylose (Fig. 4b) underpin that this improvement can be attributed to the de-repression of native glycolysis enzymes.

The reduced lag phase was also observed for PD584 grown on hexose substrates glucose and fructose (Supplementary Fig. 2). The high growth rate of PD584 on fructose (μmax = 0.30 ± 0.00 h$^{-1}$), which is similarly to xylose metabolized through Pgi, Zwf, Pgl, and the ED pathway[48], indicated that these reactions are not limiting xylose catabolism. However, the growth rate of PD584 on xylose increased only modestly (by ~18%) compared to PD310 (Table 2) and we thus sought additional targets to eliminate further bottlenecks in the metabolism.

The most evident differences between the distribution of fluxes in the EDEMP cycle in FBA and MFA were within the PPP. During growth on native sugar substrates such as glucose or fructose, the role of PPP in *P. putida* is rather complementary and predominantly anabolic[40]. Periplasmic glucose oxidation is the preferred route for glucose uptake and therefore only a smaller fraction of carbon (~20–50%) flows through Zwf and Pgl and even much less (~1–10%) is directed to the non-oxidative branch of the PPP to provide metabolic precursors for nucleotides and some amino acids[33,35,49]. This changes during the growth of recombinant *P. putida* on xylose for which PPP is the metabolic entry point. The situation is reminiscent of the exposure of *P. putida* to oxidative stress in which case activities of Zwf and Gnd

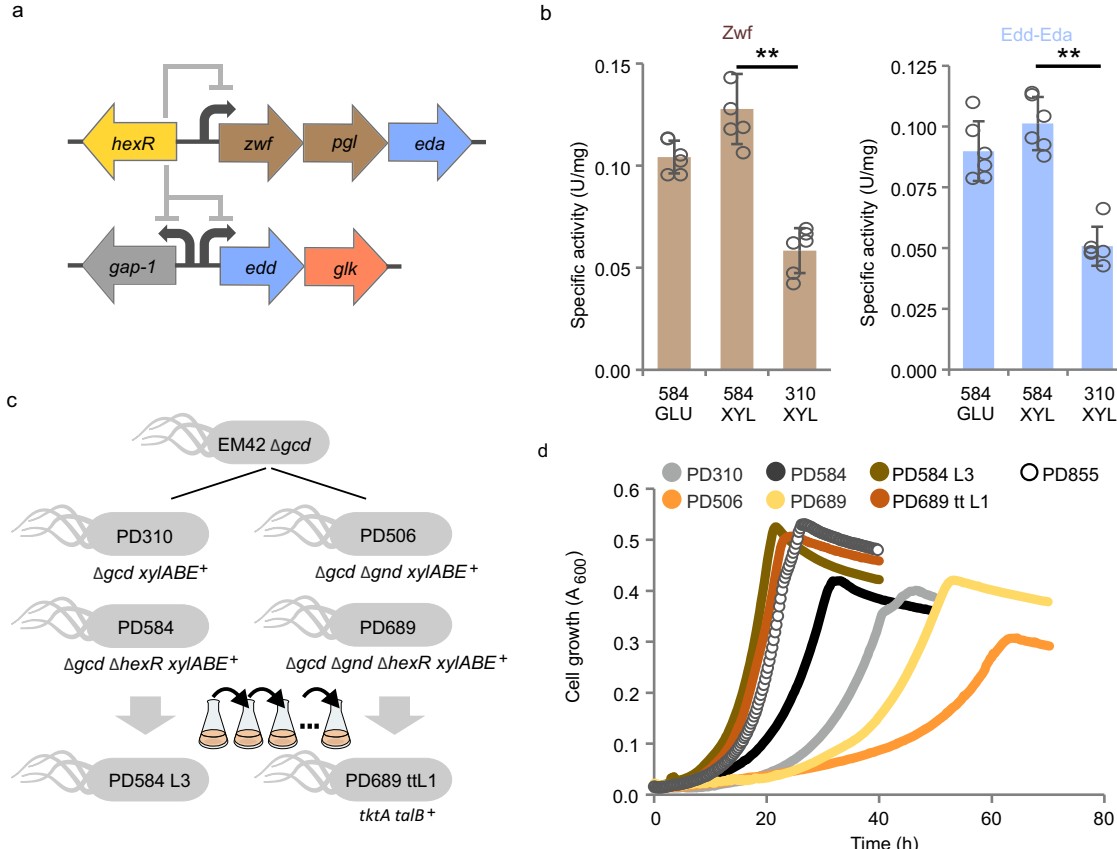

**Fig. 4 | Synthetically-primed adaptation of *Pseudomonas putida* to D-xylose.**
**a** Genetic organization of relevant genes in operons regulated by HexR transcriptional regulator. The elements in this scheme are not drawn to scale. Abbreviations used are the same as in Fig. 1. **b** Specific activity of Zwf and Edd-Eda measured in cell-free extracts from PD310 or PD584 cells grown in M9 medium with 2 g L$^{-1}$ glucose (GLU), or in M9 medium with 2 g L$^{-1}$ xylose (XYL). Data are shown as mean ± standard deviation from six ($n = 6$) biological replicates. Asterisks (**) denote statistically significant difference between two means at $p < 0.01$ calculated using two-tailed Student $t$ test ($p$ values = $2.25 \times 10^{-5}$ for Zwf and $6.81 \times 10^{-6}$ for Edd-

Eda). **c** Pedigree of *P. putida* mutant strains used in this study. Abbreviations are the same as in the previous figures. The graphical scheme shows genes deleted and introduced rationally in these strains and highlights two lineages of mutants with (PD310, PD584, PD584 L3) and without (PD506, PD689, PD689 tt L1) *gnd* gene. **d** Growth of PD310, PD506, PD584, PD584 L3, PD689, PD689 tt L1, and reverse engineered PD855 in M9 medium with 2 g L$^{-1}$ D-xylose in a 48-well microplate. Data are shown as mean from six ($n = 6$) biological replicates. Error bars are omitted for clarity. Source data are provided as a Source Data file.

increase multiple times to generate reducing equivalents for the elimination of reactive oxygen species (ROS)[40]. This adaptation has no significant effect on the growth rate and indicates that the bacterium has the capacity to adjust its PPP in favor of non-native pentose metabolism. However, during the growth on xylose, all carbon enters the PPP at the point of X5P, not G6P, as is the case during *P. putida´s* physiological response to ROS. Hence, the suboptimal activity of the non-oxidative branch of PPP might still be limiting xylose metabolism.

Elmore and co-workers (2020) accelerated the growth of KT2440 *xylABE*⁺ on xylose by enhancing it with additional transketolase (*tktA*) and transaldolase (*talB*) genes from *E. coli*, which grows well on pentoses[19]. A similar approach has been successfully employed for other bacteria, e.g., *Zymomonas mobilis*[50]. Guided by these studies and our flux analysis results, we decided to modulate the PPP in strain PD584 to further improve xylose utilization. To mimic the FBA scenario with zero flux through the Gnd reaction and to test the carbon-saving potential of this setup, we prepared a *P. putida* strain designated PD689 with deletions of *gcd*, *hexR*, and *gnd* and harboring the pSEVA2213_*xylABE* plasmid (Table 1, Fig. 4c, Supplementary Fig. 3). PD689 demonstrated a lower growth rate and 4-fold longer lag phase compared to the PD584 reference (Table 2, Fig. 4d), which confirmed that the null mutation of *gnd* decelerates growth but is not fatal for xylose utilization by *P. putida* (Fig. 3). We then integrated an expression cassette bearing either *talB-tktA* or *talB-tktA-rpe-rpiA* synthetic operon assembled from *E.*

*coli* genes into the genome of PD584 and PD698. We presumed that the activity of Rpe epimerase and RpiA isomerase (Fig. 2), whose genes were included in the second variant of the operon, could provide a higher carbon pull, support the conversion of X5P to R5P, and thus replenish metabolites depending on the Gnd reaction.

The expression cassettes were inserted randomly into the host´s chromosome using mini-Tn5 delivery plasmid pBAMD1-4[51]. This allowed for a chromosome position effect and selection of transformants with optimal gene expression. The cassettes were complemented with the constitutive P$_{EM7}$ promoter ensuring transcription at various integration sites[28]. Following the transformation with pBAMD1-4 constructs and a 4-5 day selection period, strains PD584 and PD689 were subjected to ALE on xylose for two weeks (Fig. 4c, Fig. 5, Methods)[8,9,19-22,52].

From an array of candidates isolated during ALE (Supplementary Fig. 4a, b), we selected three clones that demonstrated the fastest growth on xylose in shake flasks and reached OD$_{600}$ > 3.5 within 24 h. The three clones, designated PD584 L3, PD584 tt L3, and PD689 tt L1, were isolates from the end of the evolutionary experiment (~60-70 generations). Interestingly, mutant PD584 L3 was isolated from the control culture (PD584 transformed with pure water instead of any of the pBAMD1-4 constructs), and strains PD584 tt L3 and PD689 tt L1 from cultures of transformants with the inserted pBAMD1-4_*talB-tktA* construct. No isolates with the integrated *talB-tktA-rpe-rpiA* operon outperformed the other strains in shake flasks (Supplementary Fig. 4c).

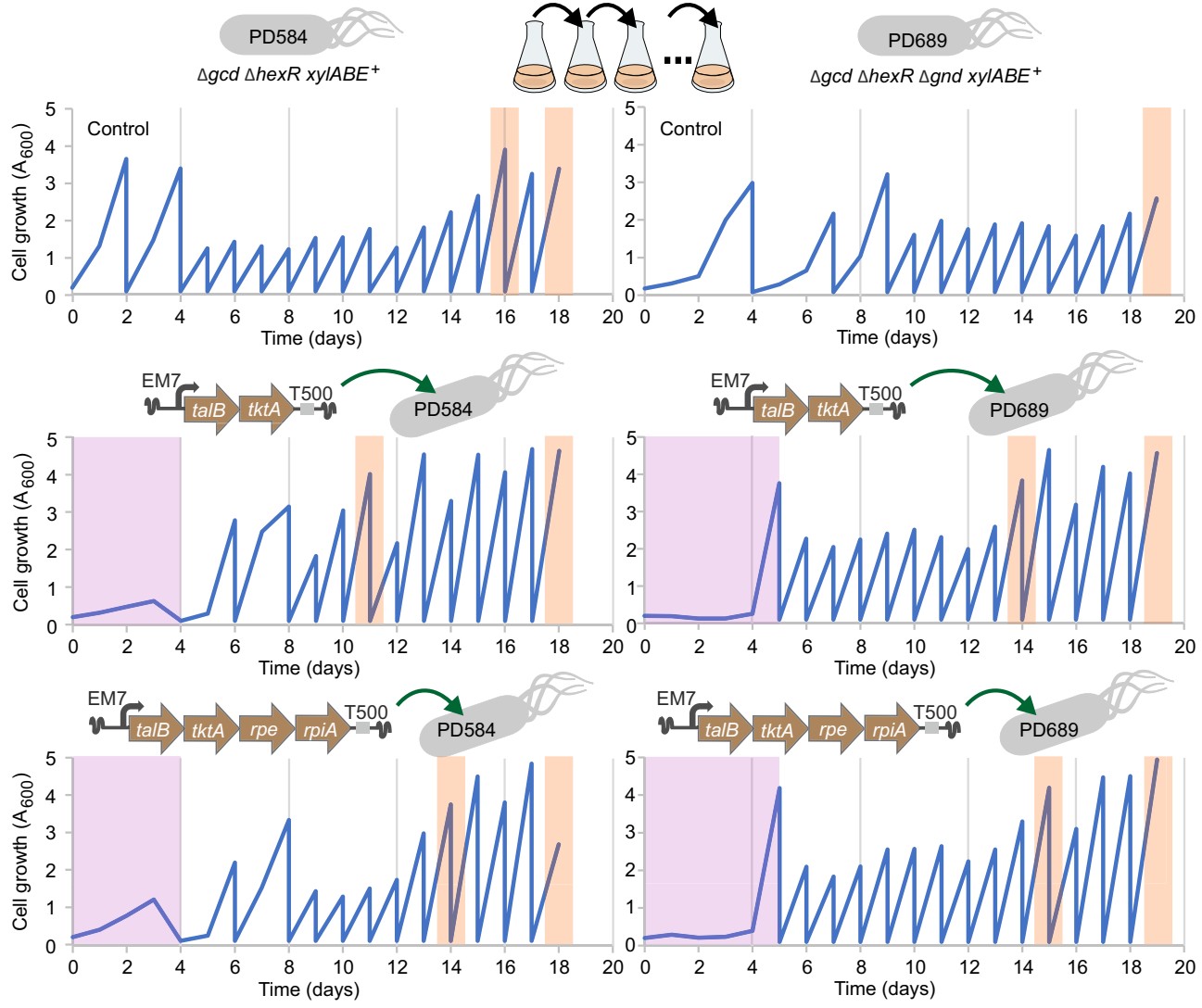

**Fig. 5 | Adaptive laboratory evolution (ALE) on xylose of *Pseudomonas putida* PD584 and PD689 strains with and without integrated synthetic operons bearing pentose phosphate pathway (PPP) genes from *Escherichia coli*.** Cells were cultured in 20 mL of M9 medium with 5 g L$^{-1}$ D-xylose and kanamycin and passaged in time intervals indicated in the graphs (upper two graphs depict ALE of PD584 and PD689 controls without implanted PPP genes and the lower four graphs depict ALE of PD584 and PD689 transformants after insertion of pBAMD1–4 plasmid constructs with PPP genes). The initial period where PD584 and PD689 transformants were cultured in M9 medium with xylose and two antibiotics (kanamycin, streptomycin) to select for the integration of *tktA-talB* and *tktA-talB-rpe-rpiA* cassettes is indicated by pink shading. ALE of cultures in M9 medium with xylose and a single antibiotic kanamycin followed with regular culture transfers. Days in which samples of the cultures were withdrawn to prepare glycerol stocks and select individual clones are highlighted by orange shading. The first withdrawal occurred when the turbidity of a given culture (OD$_{600}$) first reached the value of $\geq 3.5$ within 24 h of growth (the only exception was the control culture of PD689 which did not pass this threshold during the evolutionary experiment). The elements in the operon schemes are not drawn to scale. EM7 constitutive promoter, T500 transcriptional terminator. The *talB*, *tktA*, *rpe*, and *rpiA* denote genes that encode *E. coli* transaldolase B, transketolase A, ribulose-phosphate 3-epimerase, and ribose-5-phosphate isomerase, respectively. Source data are provided as a Source Data file.

It is plausible that the *E. coli rpe* and *rpiA* genes were not needed for the improved xylose utilization in *P. putida*.

The ALE and subsequent screening experiments showed that the evolution of the PD584 strain can provide variants with substantially accelerated growth on xylose even without supplementation of any additional exogenous genes. That is particularly intriguing considering other recent studies that endowed *P. putida* with *E. coli* genes for better growth on xylose[19,21]. It demonstrates that a similar outcome can be achieved with fewer engineering steps. A 3-week control ALE of *P. putida* PD310 did not result in enhanced growth on xylose (Supplementary Fig. 5). This suggests that the *hexR* deletion in PD584 and PD689 is an important factor for the enhanced carbon passage through the native glycolysis reactions (namely, the Zwf-Pgl-Edd-Eda part of the EDEMP cycle) and fast evolution of these strains[12]. Growth assays confirmed that all three evolved strains utilize xylose more

efficiently than their ancestors (Table 2, Fig. 4d, Supplementary Fig. 6). Remarkable was the 5-fold reduction of the lag phase of PD698 tt L1 compared to PD689 (Table 2, Fig. 4d). PD689 tt L1 utilized xylose as efficiently as PD584 L3, which, however, originated from the much better-growing ancestor PD584 (Table 2, Fig. 4d).

Additional cultivation experiments with glucose (Supplementary Fig. 7a, Supplementary Note 1) revealed a partially negative effect of metabolic adjustments on the growth rate and lag phase of PD584 tt L3 and PD689 tt L1 on this substrate. However, both PD689 tt L1 and PD584 L3 maintained the previously reported ability of the ancestral strain PD310 to co-utilize glucose and xylose (Supplementary Fig. 7b)[18].

### Unveiling the causes of the improved xylose utilization
We combined proteomic analysis with whole-genome sequencing to correlate proteome changes with genomic alterations in *P. putida*

PD310, PD584, PD584 L3, and PD689 tt L1 grown on xylose. Strain PD584 tt L3 was excluded from these characterizations because its growth on glucose was impaired (Supplementary Fig. 7a) and sequencing revealed that, in contrast to PD689 tt L1, the *talB-tktA* cassette was not incorporated into its genome, and therefore its genotype was similar to strain PD584 L3 (Supplementary Note 2 and Supplementary Table 1).

Altogether, 3981 proteins (92 duplicities removed) were uniquely detected in the four strains (Supplementary Data 2). The proteomic analysis showed relatively subtle differences between PD584 and PD310 (118 downregulated and 87 upregulated proteins, $\log_2$fold change >1.0, $p < 0.05$, Supplementary Fig. 8) and between the evolved strain PD584 L3 and its template PD584 (83 downregulated and 58 upregulated proteins, Supplementary Fig. 8), while the proteomes of the two evolved strains PD689 tt L1 and PD584 L3 varied in 667 proteins (376 downregulated and 292 upregulated, Supplementary Fig. 8). Visualizing the changed protein abundances in the *P. putida* central carbon metabolism map verified the effects of our targeted engineering interventions and disclosed additional mutations selected in the ALE experiment (Fig. 6).

Sequence analysis revealed various polymorphisms affecting the sequence of the encoded proteins in parental strains PD584 and PD689 (Supplementary Data 3, 4). In the evolved strains, changes represented by de novo mutations were significantly more prevalent than fixed polymorphisms. Due to the detected genomic changes, the chromosomes of PD584 L3 and PD689 tt L1 showed 99.8% and 99.9% pairwise identity, respectively, to their parental strains (using whole-genome alignment by Mauve Plugin in Geneious Prime, Supplementary Data 3, 4). Given the known variability of pseudomonad genomes, such changes are not surprising[53]. Numerous mutations accumulated in PP_0168 (locus tag PED37_RS00910 in PD584) encoding the large adhesive protein LapA responsible for biofilm formation[54]. Mutations in biofilm genes can occur in response to stress or selection of planktonic cells during multiple transfers in ALE experiments, as was observed previously[47,55]. However, we did not observe a significant difference in biofilm forming ability between PD584 L3, PD689 tt L1, and PD584 used as a control (Supplementary Fig. 9 and Supplementary Method 2).

Importantly, we identified an intriguing alteration in the genomes of PD584 and PD584 L3– a multiplication of a large region (118,079 bp), discovered by an approximately 6-fold higher sequencing coverage from locus PP_2114 to PP_2219 (Fig. 7a, Supplementary Table 2, Supplementary Data 5). Further investigation revealed that this multiplication was already present in the previously published strain PD310[18], but was not present in its EM42 Δ*gcd* template (Supplementary Table 2). Freshly prepared EM42 Δ*gcd* pSEVA2213_*xylABE* exhibited a similar growth rate as the original PD310 but the fresh clones passed through a very long lag phase (~40 h of slow linear growth) before growing exponentially (Table 2, Fig. 7b). This experiment demonstrated a positive effect of the multiplication on xylose utilization. The multiplication was maintained in the whole lineage of PD310, PD584, and PD584 L3 strains (Fig. 4c) but was absent in PD689 tt L1 and its ancestor PD689.

As its two border open reading frames (ORFs) encode transposases, we presume that the multiplication occurred via repeated transposition events. The multiplied region includes 108 CDS of which 9 encode hypothetical proteins. The remaining CDS encode transporters or their subunit (3 genes), transcriptional regulators (7 genes), or enzymes (53 genes) (Supplementary Data 5). Notably, the latter set includes the gene of transaldolase Tal (PP_2168). We cloned the *tal* gene into the expression plasmid pSEVA438 (Supplementary Method 3, 4, Supplementary Data 6) and inserted the resulting construct into the strain EM42 Δ*gcd* pSEVA2213_*xylABE*. Growth comparison of this strain on xylose with control (EM42 Δ*gcd* pSEVA2213_*xylABE* + pSEVA438) demonstrated that overexpression of

the *tal* gene alone ensured an improved phenotype, which in the case of PD310 was made possible by amplification of the entire 118 kb segment including *tal* (Fig. 7c).

We hypothesized that the transposition and segment multiplication in strains PD310 and PD584 occurred during their re-streaking on agar plates with xylose, which we performed to check the desired phenotype before glycerol stock preparation. To verify this, we streaked several clones of the freshly prepared EM42 Δ*gcd* pSEVA2213_*xylABE* strain without multiplication on a xylose agar plate. When the clones were re-streaked on a fresh plate and then cultivated in liquid microplate cultures, we indeed observed faster growth, comparable to the PD310 control (Supplementary Fig. 10). This growth acceleration can thus be attributed to the replication of the large genomic segment with the *tal* gene, which was identified in the chromosomes of all four clones (Supplementary Table 2).

Proteome comparison of PD584 and PD310 found that the *hexR* deletion in PD584 manifested in increased abundances of the enzymes ZwfA, Pgl, Edd, Eda, and Gap (Fig. 6a) as expected. A potential redox imbalance caused by higher activity of de-repressed dehydrogenases Zwf and Gap could explain the apparent upregulation of glyoxylate shunt enzymes isocitrate lyase AceA (PP_4116) and malate synthase GlcB (PP_0356), as discussed in the first part of the Results and discussion section. The modestly increased abundance of Tal transaldolase (PP_2168) (Fig. 6a, Supplementary Data 5) can be attributed to a higher copy number of the multiplied PP_2114−PP_2219 segment in the chromosome of PD584 (six copies) compared to PD310 (four copies, Supplementary Table 2).

Inspection of downregulated and upregulated proteins in the central carbon metabolism of PD584 L3 compared to PD584 revealed only minute changes including a small decrease in the quantity of ZwfA in the EDEMP cycle, AceA in the glyoxylate shunt, and malate dehydrogenase Mdh (PP_0654) in the TCA cycle (Fig. 6b). The multiplication could not be responsible for the improved phenotype of the evolved mutant PD584 L3 as the same number of copies was detected in both strains (Supplementary Table 2), and other genomic changes that we identified and were able to interpret did not clearly explain the improved phenotype either. Finally, sequencing of the pSEVA2213_*xylABE* plasmid revealed a perfect 32 bp duplication upstream of the *xylA* gene in PD584 L3 (Fig. 7f). The duplication encompassed the synthetic Shine−Dalgarno sequence (RBS), the ATG start codon and the following eight nucleotides of the *xylA* gene. Analysis of the resulting mRNA sequence with RBS Calculator[30] revealed that the RBS in the duplication with a predicted translation initiation rate of 310 a.u. became the strongest RBS upstream of the *xylA* gene while the strength of the original *xylA* RBS was reduced 10-fold from 292 to only 29 a.u. It is plausible that the effect of the duplication lies in the emergence of a mechanism similar to a translational coupler, that is, a region downstream of the promoter that encodes a short leading peptide stabilizing translation of the downstream gene[56]. However, no such peptide was identified in the PD584 L3 proteome, so the specific molecular effect of the duplication remains to be elucidated (Methods).

Since an increase in exogenous XylA abundance was confirmed on the protein level (Fig. 6b), it was probable that the duplication caused higher *xylA* expression. To verify the hypothesized effect of the duplication on xylose utilization by *P. putida*, PD584 L3 was deprived of the plasmid by sub-culturing in rich LB medium without antibiotics and then transformed either with the same mutated plasmid or with the original pSEVA2213_*xylABE*. The strain with the mutated plasmid grew equally well on xylose compared to PD584 L3 ($\mu = 0.21 \pm 0.00\ \text{h}^{-1}$), while the strain transformed with the original plasmid grew 33% slower ($\mu = 0.14 \pm 0.00\ \text{h}^{-1}$, Supplementary Fig. 11a). Similarly, when the mutant plasmid from PD584 L3 was inserted into PD584 strain devoid of its own pSEVA2213_*xylABE* construct, the resulting strain showed faster growth on xylose than

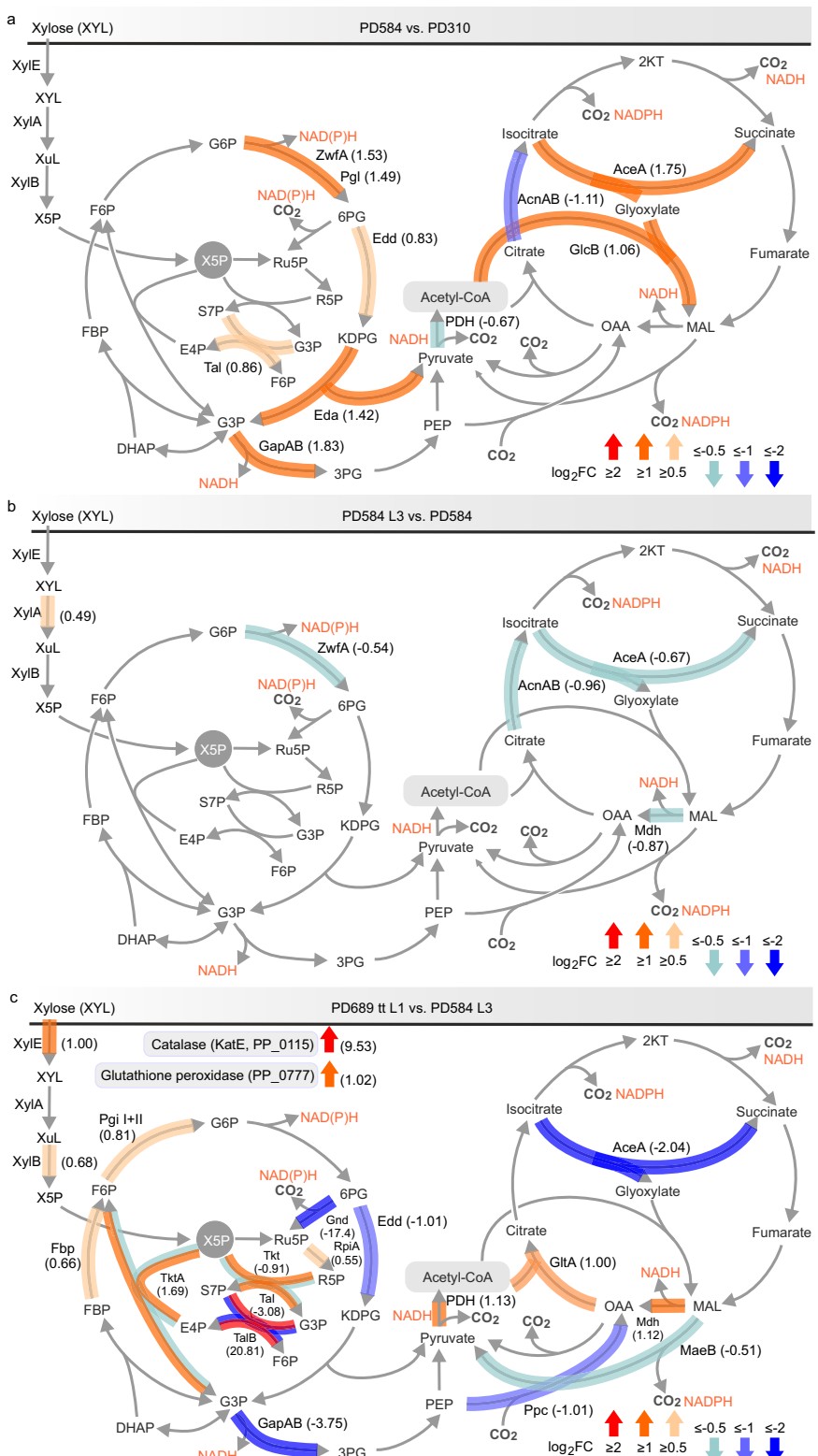

**Fig. 6 | Changes in protein abundances in the upper xylose pathway and central carbon metabolism in selected *Pseudomonas putida* strains.** Changes in protein abundances for (**a**) strain PD584 compared to PD310, (**b**) PD584 L3 compared to PD584, and (**c**) PD689 tt L1 compared to PD584 L3. Normalized and imputed protein intensities were used for differential expression using LIMMA statistical test. The figures show log₂(fold change, FC) values for significantly differentially expressed proteins (adj. *p* ≤ 0.05). The *p* values adjustment on multiple hypothesis testing was done using Benjamini & Hochberg method. The metabolic map and abbreviations used are the same as in Fig. 2. Please note that Tal and Tkt represent native *Pseudomonas putida* transaldolase (PP_2168) and transketolase (PP_4965), respectively, while TalB (KEGG ID: JW0007) and TktA (KEGG ID: JW5478) stand for the respective enzymes from *Escherichia coli*. Note that proteins with less significant changes in abundance (log₂FC ≥ 0.5 or ≤ −0.5) were also visualized. Source data are provided as a Source Data file.

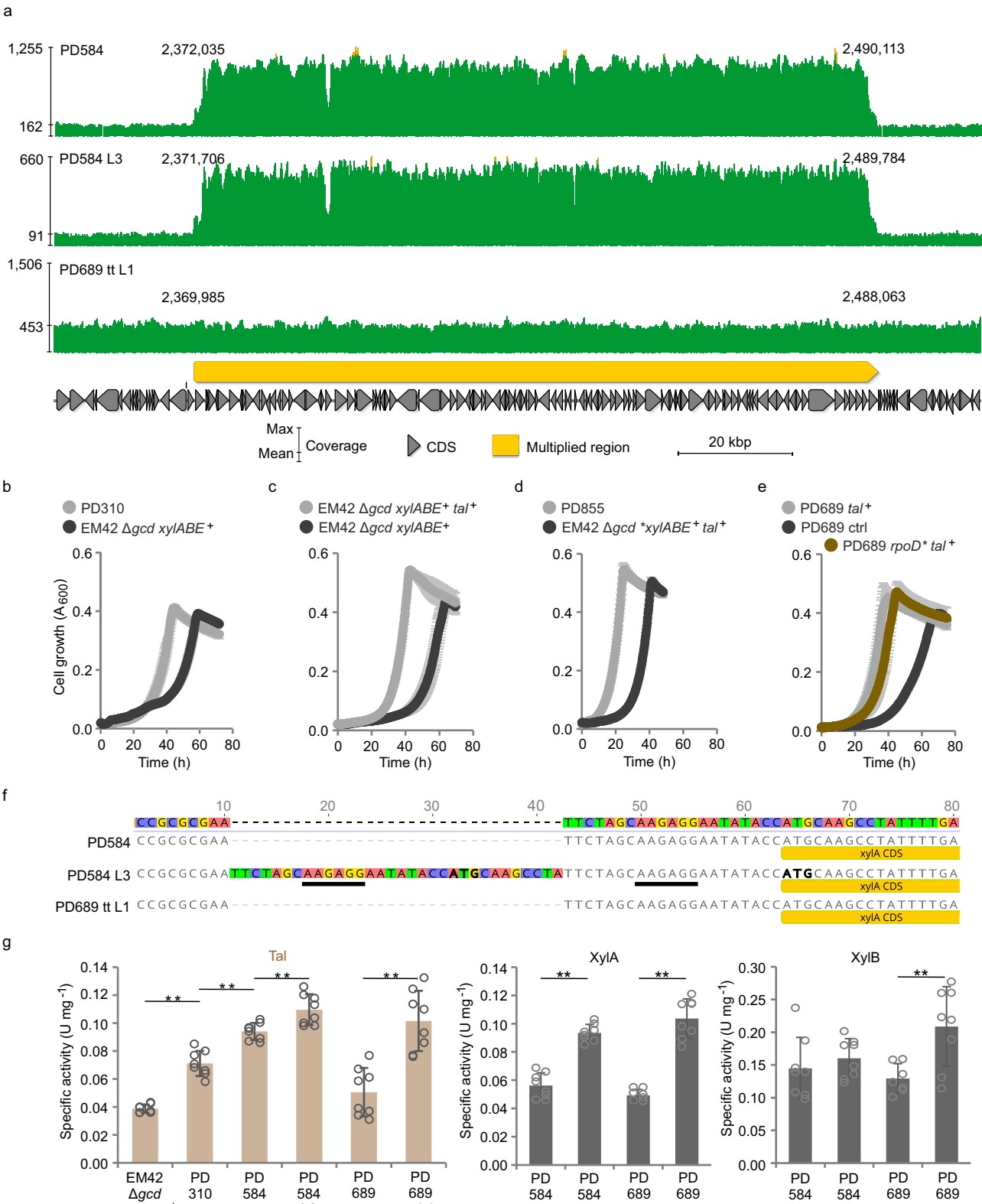

PD584 (Supplementary Fig. 11b). Interestingly, when we introduced the mutated plasmid into plasmid-less PD689 tt L1, the resulting strain did not grow better than PD689 tt L1 (Supplementary Fig. 11c). We thus demonstrated that the mutated pSEVA2213_xylABE plasmid is partially responsible for the improved growth of PD584 L3 on xylose but does not give any additional advantage to PD689 tt L1. To test for an additive effect of the duplication upstream of the xylA

gene and tal overexpression, we inserted the mutated pSE-VA2213_xylABE plasmid and pSEVA438_tal into the prepared strain EM42 Δgcd ΔhexR (PD580, Table 1) without multiplication in its chromosome (Supplementary Tables 1, 2). The growth parameters of the resulting reverse engineered strain PD855 are close to those of PD584 L3 (Tables 1, 2, Fig. 4d, Supplementary Fig. 6). Comparison of the growth of PD855 (lag phase 4.94 ± 0.39 h) with the same strain

**Fig. 7 | Characterization of engineered and evolved *Pseudomonas putida* EM42 strains. a** Multiplied (6x) region identified from the whole-genome sequencing data in the chromosome of PD584 and PD584 L3. **b** Comparison of the growth of *P. putida* PD310 to *P. putida* EM42 Δ*gcd* pSEVA2213_*xylABE*, (**c**) *P. putida* EM42 Δ*gcd* pSEVA2213_*xylABE* + pSEVA438_*tal* to *P. putida* EM42 Δ*gcd* pSE-VA2213_*xylABE* + pSEVA438, (**d**) *P. putida* PD855 to *P. putida* EM42 Δ*gcd* pSE-VA2213_**xylABE* + pSEVA438_*tal* (*denotes 32 bp duplication in plasmid from PD584 L3), and (**e**) PD689 pSEVA438_*tal* to PD689 pSEVA438 (ctrl) and PD689 *rpoD**pSEVA438_*tal* (*denotes *rpoD* gene with Ser552Pro mutation). Cultures were grown in a 48-well microplate with M9 minimal salts medium and 2 g L$^{-1}$ xylose. 3-Methylbenzoate (25 μM) was added to the cultures of cells carrying pSEVA438 or pSEVA438_*tal* plasmid at time 0 h. PD310 contains the multiplication of the -118 kbp segment in its chromosome. Data are shown as mean ± standard deviation from six

($n = 6$) biological replicates. **f** 32 bp duplication identified upstream of the *xylA* gene in pSEVA2213_*xylABE* plasmid isolated from PD584 L3. The duplication includes a synthetic RBS AAGAGG (underlined), ATG codon (in bold) followed by eight nucleotides from 5´-end of the *xylA* gene. The coverage graphs in (**a**) as well as the sequence alignment in (**f**) were generated by Geneious Prime 2022.2.2. **g** The specific activity of transaldolase, xylose isomerase, and xylulokinase determined in cell-free extracts of *P. putida* strains. Columns represent means ± standard deviations calculated from eight ($n = 8$) biological replicates from two independent experiments. Asterisks denote the significance of the difference between the two means at $p < 0.01$ calculated using two-tailed Student *t* test ($p$ values from left to right = $7.17 \times 10^{-6}$, $5.99 \times 10^{-5}$, $5.12 \times 10^{-3}$, $1.53 \times 10^{-4}$, $3.84 \times 10^{-7}$, $4.54 \times 10^{-6}$, $6.95 \times 10^{-3}$). Source data are provided as a Source Data file.

but with functional *hexR* (lag phase 20.80 ± 0.31, Fig. 7d) also confirmed the importance of the *hexR* deletion.

Comparison of proteomes of two superior strains PD689 tt L1 and PD584 L3 revealed many intriguing differences and confirmed that PD689 tt L1 with the deleted 6-phosphogluconate dehydrogenase gene *gnd* followed a different evolutionary path to attain an improved xylose utilization phenotype (Fig. 6c, Supplementary Fig. 8). The proteomic data further support the existence of a relationship between redox balance changes (here due to the altered flux through the Gnd reaction) and glyoxylate shunt activity[26,34]. Isocitrate lyase AceA was significantly downregulated in *gnd*$^-$ PD689 tt L1 (Fig. 6c), while upregulation of the pyruvate dehydrogenase complex, citrate synthase GltA (PP_4194), and malate dehydrogenase Mdh supported TCA cycle activity.

We expected that some of the lower expressed proteins in PD689 tt L1 were encoded by the chromosomal region (PP_2114−PP_2219), multiplied in PD584 L3 but not in PD689 tt L1 (Supplementary Data 7). Indeed, in PD689 tt L1, native Tal and Tkt were downregulated compared to PD584 L3 while Rpe and RpiA enzymes showed no or modest change in quantity (Fig. 6c). In turn, the presence of the exogenous *talB-tktA* cassette in PD689 tt L1 supplied the observed upregulation of the transaldolase and transketolase enzymes in this strain. This result again confirmed that transaldolase is a key PPP enzyme whose activity is one of the major determinants of the xylose utilization rate in *P. putida*. In contrast to PD310, PD584, and PD584 L3, which benefited from several copies of the native *tal* gene in the multiplied chromosome segment and where the presence of Gnd enabled carbon cycling and replenishment of the ribulose 5-phosphate and ribose 5-phosphate pool, the *gnd*$^-$ mutant PD689 tt L1 enhanced the efficiency of the non-oxidative PPP by incorporating the enzymes transplanted from *E. coli*. This is further supported by the measurements of transaldolase activity in six *P. putida* strains discussed in this study (Fig. 7g). Utilization of the exogenous TalB in PD689 tt L1 appears to be an adaptive response, necessitated by the absence of the genomic multiplication and the Gnd enzyme in its parental strain PD689. In mutants with *gnd* removed based on the computer-aided design, the absence of 6-phosphogluconate dehydrogenase activity had to be compensated. The implanted TalB could provide the necessary pull effect to promote a flow of carbon through the preceding PPP reactions starting with xylulose 5-phosphate[7]. Moreover, the adoption of two exogenous genes could be a more economical solution in terms of cellular resource use than the amplification of a large genomic region.

Another essential element of the engineered xylose metabolism in *P. putida* EM42 involves the expression of *xylA*, *xylB*, and *xylE* genes of the synthetic operon. We measured activities of XylA and XylB in PD584, PD584 L3, PD689, and PD689 tt L1 grown in M9 medium with xylose and found that the XylA activity in PD584 L3 with the duplication upstream of the *xylA* gene was almost twice as high as the activity measured in the PD584 strain (Fig. 7g). To our surprise, increased activity as well as protein abundance of XylA was determined also for PD689 tt L1 (Figs. 6c, 7g). Moreover, XylB and XylE were upregulated in

this evolved strain (Figs. 6c, 7g). In the study of Elmore and co-workers, higher expression of the *xylE* gene caused by a mutation in its promoter was identified as the major determinant of improved xylose utilization by engineered *P. putida* KT2440[19], but in our case, no mutation was pinpointed in the pSEVA2213_*xylABE* plasmid isolated from PD689 tt L1. An additional experiment excluded the possibility of an increased copy number of the pSEVA2213_*xylABE* plasmid in this strain when compared to its ancestor PD689 (Supplementary Fig. 12). Upregulation of XylE, XylA, and XylB in this strain thus signified a higher expression of the whole synthetic *xylABE* operon. We hypothesized that the explanation lays in the identified missense mutation Ser552→Pro552 in the *rpoD* gene (PP_0387) encoding the RNA polymerase sigma factor σ$^{70}$, which directs the binding of the RNA polymerase complex to promoters of housekeeping genes. The mutation leads to a change in one of the α−helices of the σ4 domain interacting with the −35 promoter element[57]. Mutations in the transcription machinery and specifically in *rpo* genes were previously described in several *E. coli* strains exposed to ALE and were identified as a prominent means of metabolism re-wiring and improved resilience against environmental perturbations[58–60]. It was conceivable that mutations in *rpo* genes could induce similar effects in *P. putida*[52]. We argued that the mutation could have increased the affinity of the *P. putida* RNA polymerase complex to the synthetic P$_{EM7}$ promoter, among others, and thus enhanced the expression of the exogenous *E. coli* genes *xylA*, *xylB*, and *xylE*[61–63]. Such an effect was confirmed by the results of a complementary experiment in which PD689 tt L1 strain with pSEVA2213_P$_{EM7}$_*xylABE* plasmid exchanged for pSEVA2213_P$_{EM7}$_*gfp* produced significantly more green fluorescent protein (GFP) during its growth on LB medium (by 77%) than the parent strain PD689 bearing the same plasmid (Supplementary Fig. 13). The same effect of the mutated *rpoD* was also confirmed with different genetic background of the PD580 strain having the transplanted Ser552Pro mutation (Supplementary Fig. 13, Supplementary Method 5). Interestingly, reverse engineering of the Ser552Pro mutation into the PD689 parent with added pSEVA438_*tal* plasmid did not result in a strain with growth similar to PD689 tt L1 (μ = 0.13 ± 0.00 h$^{-1}$, lag phase = 14.1 ± 0.4 h), although overexpression of *tal* alone improved the growth of PD689 (Fig. 7e). The effect of *rpoD* mutation on xylose utilization is probably more complex and requires precise orchestration with other changes that emerged in the evolved PD689 tt L1 (Supplementary Data 4). Higher expression of the whole *xylABE* operon and the adoption of exogenous TalB and TktA enzymes do not need to have only positive effects on PD689 tt L1. The substantially higher quantity of ROS-reducing enzymes – catalase KatE and glutathione peroxidase – in PD689 tt L1, when compared to PD584 L3 (Fig. 6c), can reflect metabolic perturbations of this strain. The stress can stem from the overproduction of the exogenous transporter XylE[64] or the implanted *E. coli* pathways and fluxes re-routed due to the *gnd* deletion. The metabolic perturbations of PD689 tt L1 were confirmed in the subsequent assay in which we challenged the five recombinant strains with an extra biosynthetic task on top of the non-native substrate utilization

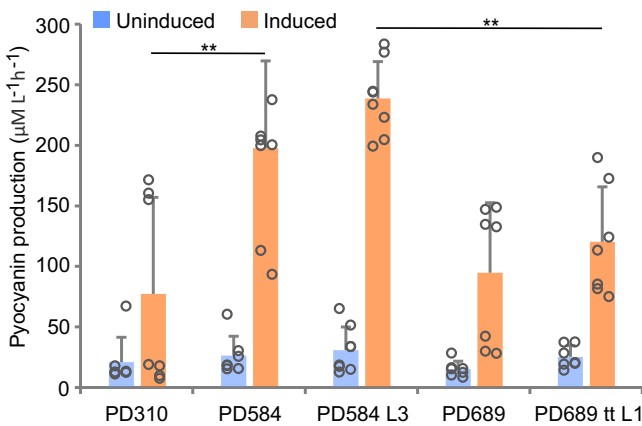

**Fig. 8 | Production of pyocyanin by engineered *Pseudomonas putida* strains.** Pyocyanin concentration in cell cultures was measured after 24 h. Columns represent means ± standard deviations calculated from at least seven (*n* = 7) biological replicates from two independent experiments. Asterisks (\*\*) denote statistically significant difference between two means at *p* < 0.01 calculated using two-tailed Student *t* test (*p* values = $9.78 \times 10^{-3}$ and $1.47 \times 10^{-4}$). Source data are provided as a Source Data file.

(Fig. 8). We transformed the strains with an additional plasmid carrying a biosynthetic pathway for the heterocyclic pigment pyocyanin, encoded by nine genes *phzA1-G1*, *phzM*, and *phzS* (Supplementary Fig. 14, Supplementary Data 6,)[22,65]. Pyocyanin exhibits strong redox activity (NAD(P)H oxidation and ROS generation) and has been studied for its antimicrobial properties[66,67]. The titer of pyocyanin produced after 24 h was significantly higher (*p* < 0.01) in cultures of PD584 and the evolved PD584 L3 than in cultures of the slower growing strains PD310 and PD689 (Fig. 8, Supplementary Method 6). PD689 tt L1, in contrast, surpassed PD310 and PD689 neither in pyocyanin production nor in the growth after pyocyanin pathway induction. The pyocyanin experiment demonstrates that accelerated substrate utilization by tailored strains does not necessarily secure enhanced synthesis of a selected bioproduct. For parallel improvement of substrate utilization and bioproduction capacity, the adaptive evolution of a given host should directly couple cell fitness to the synthesis of a desired chemical[4,47,68].

In conclusion, our study leveraged *P. putida's* remarkable genomic and metabolic plasticity to adapt the strain to a non-native substrate. The adaptation involved a combination of rational engineering and evolutionary events, including a large genome re-arrangement, a smaller duplication, and single nucleotide substitutions. In the lineage of strains PD310 and PD584 the multiplication of the genomic segment PP_2114−PP_2219 with the *tal* gene and the *hexR* deletion were pivotal in overcoming metabolic bottlenecks preventing faster growth on xylose. These changes facilitated further adaptation of the bacterium to xylose, leading to PD584 L3. The transaldolase gene overexpression was also important for the improved growth of PD689 tt L1. Both PD689 tt L1 and PD584 L3 strains demonstrated a different strategy for balancing gene expression in the *xylABE* operon. Strain PD584 L3 finetuned the expression of *xylA* by embossing the gene with a variant of translational coupler. In contrast, the adaptation of PD689 tt L1 occurred through a more "brute-force" approach – a missense mutation in the sigma factor RpoD, which enabled, among others, enhanced expression of the whole synthetic *xylABE* operon. The optimal nesting of the new catabolic route in the host's biochemical network required substantial adjustments and finetuning on metabolic and regulatory levels, which were informed by existing knowledge but not entirely predictable through metabolic models alone[10]. The resulting *P. putida* strains PD584 L3 and PD689 tt L1 occupy discrete local optima on the fitness landscape, achieved through different evolutionary paths. In

agreement with other recently published works, *P. putida* KT2440 emerged as an attractive candidate for evolutionary experiments that can be primed by rational metabolic engineering and introduced synthetic genetic devices[4,69]. Our approach successfully delivered *P. putida* strains with a doubled growth rate and substantially reduced lag phase on xylose, including a reverse engineered strain PD855 with only two targeted deletions (Δ*gcd* Δ*hexR*) and three exogenous genes (*xylABE*), positioning them as valuable templates for biotechnological valorization of xylose or its co-valorization with glucose. The findings presented in this study are instrumental for future attempts to exploit semi-synthetic xylose metabolism in *P. putida* KT2440 and its derivatives for biotechnological purposes as well as for the understanding of bacterial adaptation to new substrates.

## Methods

### Bacterial strains and conditions of routine cultures

Bacterial strains used in this study are listed in Table 1 (*P. putida* strains) and Supplementary Table 3 (*E. coli* strains). The strains were routinely grown in lysogeny broth (LB; 10 g L$^{-1}$ tryptone, 5 g L$^{-1}$ yeast extract, 5 g L$^{-1}$ NaCl) with agitation (350 rpm, Heidolph Unimax 1010 and Heidolph Incubator 1000; Heidolph Instruments) at 30 °C (*P. putida*) or 37 °C (*E. coli*). Antibiotics – kanamycin (Km, 50 μg mL$^{-1}$), ampicillin (Amp, 150 or 450 μg mL$^{-1}$ in *E. coli* and *P. putida* cultures, respectively), streptomycin (Sm, 50 or 60 μg mL$^{-1}$ in *E. coli* and *P. putida* cultures, respectively), chloramphenicol (Cm, 30 μg mL$^{-1}$) or gentamicin (Gm, 10 μg mL$^{-1}$) – were added to liquid or solid media for plasmid maintenance and selection. *P. putida* strains were routinely pre-cultured overnight (16 h) in 50 mL tubes with 5 mL of LB medium. All precultures were inoculated directly from cryogenic glycerol stocks stored at −70 °C and prepared from single isolated clones. For the main cultures in 250 mL Erlenmeyer flasks or 48-well microplates, overnight cultures were spun (2000 g, room temperature RT, 7 min) and washed with M9 mineral salt medium (7 g L$^{-1}$ Na$_2$HPO$_4$·7H$_2$O, 3 g L$^{-1}$ KH$_2$PO$_4$, 0.5 g L$^{-1}$ NaCl, 1 g L$^{-1}$ NH$_4$Cl$_2$, 2 mM MgSO$_4$, 100 μM CaCl$_2$, 20 μM FeSO$_4$) supplemented with 2.5 mL L$^{-1}$ trace element solution[70]. Cells were then resuspended to a starting OD$_{600}$ of 0.1 in 50 mL of M9 medium with Km in case of shake flask cultures or to an initial OD$_{600}$ of 0.05 (as measured in a cuvette with an optical path of 1 cm) in 600 μL of M9 medium with Km in case of 48-well plate cultures. A carbon source (D-xylose, D-glucose, or D-fructose) was added in a concentration defined in text or respective figure or table caption. Flasks cultures were incubated at 30 °C with agitation (200 rpm) using IS-971R incubated shaker (Jeio Tech) and growth was monitored by measuring the OD$_{600}$ of cultures using UV/VIS spectrophotometer Genesys 5 (Spectronic). Microplates were placed in Infinite M Plex plate reader (Tecan) and incubated at 30 °C. The OD$_{600}$ was measured every 15 min and orbital shaking (245 rpm) with 2.5 μm amplitude was applied in between measurements, linear shaking with 2.5 μm amplitude was set for 10 s before each OD measurement. To avoid condensation of water vapor on the plate lid, the inner surface was treated with a detergent solution in ethanol (0.05% v/v Triton X-100 in 20% v/v EtOH). Excess liquid was decanted and the lid was dried and UV sterilized. All solid media used (LB and M9) contained 15 g L$^{-1}$ agar. M9 solid media were supplemented with 2 g L$^{-1}$ carbon source and 50 μg mL$^{-1}$ Km.

### General cloning procedures, construction of mutant strains and plasmids

General cloning procedures are provided as Supplementary Method 3 in Supplementary Information. All plasmids used or prepared in this study are listed in Supplementary Data 6 file. Oligonucleotide primers used in this study (Supplementary Data 8) were purchased from Merck. Nucleotide sequences of constructed synthetic operons and expression cassettes are provided in Supplementary Data 9.

Preparation of deletion mutants of *P. putida* EM42: Deletion mutants EM42 Δ*gcd* Δ*gnd*, EM42 Δ*gcd* Δ*pgi*-I + II, EM42 Δ*gcd* Δ*edd*, EM42 Δ*hexR*, EM42 Δ*gcd* Δ*hexR*, and EM42 Δ*gcd* Δ*hexR* Δ*gnd* were prepared using the homologous recombination-based protocol[18,71]. Briefly, the regions of approximately 500 bp upstream and downstream of the *gnd* (PP_4043), *pgi*-I + II (PP_1808 and PP_4701), *edd* (PP_1010), and *hexR* (PP_1021) genes were PCR amplified with respective TS1F, TS1R (upstream) and TS2F, TS2R (downstream) primers (Supplementary Data 8). TS1 and TS2 fragments were joined through overlap extension or SOEing-PCR[72], and the PCR product was digested with *Eco*RI and *Bam*HI and cloned into a non-replicative pEMG plasmid. Competent *P. putida* EM42 cells were transformed with sequence-verified plasmids by electroporation. Transformants were selected on LB agar plates with Km and several co-integrates were pooled for further work. Co-integrates were transformed with the pSW-I plasmid by electroporation. Transformants were plated on LB agar plates with Amp and expression of I-*Sce*I in selected clones inoculated into 5 mL of LB was induced with 5 mM 3-methylbenzoate (3MB) for 6–16 h depending on the deletion. Induced cells were plated on LB agar plates with and without Km and clones sensitive to Km were checked for the target deletion by colony PCR using a respective TS1F/TS2R or check fw/check rv (anneal within the deleted gene) primer pair (Supplementary Data 8). *P. putida* recombinants were cured of pSW-I plasmid by several passes in LB medium lacking Amp. The preparation of *rpoD* Ser552Pro mutants of PD580 and PD689 is described in Supplementary Method 5.

Preparation of pBAMD1–4 constructs: The genes *talB* for transaldolase B (JW0007) and *tktA* for transketolase A (JW5478) were PCR amplified with their native ribosome binding sites (RBS) from *E. coli* BL21(DE3) genomic DNA using Q5 polymerase (NEB). In the second PCR step, an overhang homologous to the 5′ end of *tktA* was added to the 3′ end of *talB* to allow following SOEing-PCR and connecting the two genes in a synthetic operon. The used primers (talB_fw, talB_PCR1_rv, talB_PCR2_rv, tktA_fw, and tktA_rv) are listed in Supplementary Data 8. The construct was digested with *Sac*I and inserted into the mini-Tn5 pBAMD1–4 vector cargo site[51] downstream of the previously added constitutive P$_{EM7}$ promoter. Due to the use of a single restriction site (*Sac*I), the vector was dephosphorylated by the addition of 2 U of shrimp alkaline phosphatase (New England BioLabs) for the last 30 min of the restriction reaction. The genes *rpe* for D-ribulose-5-phosphate 3-epimerase (JW3349) and *rpiA* for ribose 5-phosphate isomerase A (JW5475) were codon optimized for *P. putida* KT2440 and complemented with synthetic RBS designed by RBS Calculator[30] to resemble the strengths of native *rpe* and *rpiA* RBS in the context of pBAMD1–4 expression cassette. The whole *rpe*-*rpiA* cassette, flanked with *Sph*I and *Not*I restriction sites, was synthesized by Eurofins Genomics. The cassette was subcloned from the delivery plasmid pEX into pBAMD1–4_EM7 or downstream of the *talB*-*tktA* genes in pBAMD1–4_EM7_*talB*-*tktA*. Chemocompetent *E. coli* CC118λπ cells or OneShot PIR1 *E. coli* cells (Thermo Fisher Scientific) were transformed with the plasmids. Cells were plated on agar plates with Sm (50 μg mL⁻¹) and grown overnight at 37 °C. The presence of the plasmid with an insert of the correct size was checked in individual clones by colony PCR and by restriction analysis of isolated plasmids. The sequences of all cloned genes were confirmed by Sanger sequencing and sequence alignments were carried out in Benchling (Alignment function MAFFT v7).

### Genomic integrations of *talB*-*tktA* and *talB*-*tktA*-rpe-*rpiA* cassettes and adaptive laboratory evolution

The previously published procedure for genomic integration using the pBAMD vector system[51] was followed with some modifications. Specifically, both *P. putida* EM42 recipient strains (PD584 double deletion mutant Δ*gcd* Δ*hexR* and PD689 triple deletion mutant Δ*gcd* Δ*hexR* Δ*gnd* both with pSEVA2213_*xylABE* plasmid) were electroporated with

the pBAMD1–4_EM7_*talB*-*tktA* or pBAMD1–4_EM7_*talB*-*tktA*-rpe-*rpiA* plasmid (100 ng). The controls were mock-transformed with 1 μL of pure miliQ water instead of plasmid DNA. After the electroporation pulse, cells were immediately resuspended in 2.5 ml of SOC medium in a 15 mL tube, and after a 5 h recovery period at 30 °C and 200 rpm, 10 μL were plated on LB agar with Sm to assess the integration efficiency. The rest of the suspension was inoculated into 20 mL of M9 medium supplemented with xylose (5 g L⁻¹), Km, and Sm (only Km was used for the controls) in a 100 mL Erlenmeyer flask and incubated at 30 °C and 200 rpm (IS-971R, Jeio Tech) overnight. The next day, the cultures were inoculated into fresh medium to an OD$_{600}$ of 0.2 and incubated for 4–5 days (time interval was longer in case of PD689 derivatives due to their negligible growth during the first four days). Then, the cells were transferred to fresh medium with xylose (5 g L⁻¹) and a single antibiotic Km to a starting OD$_{600}$ of 0.1 and passaged every 48 or every 24 h for 14 days in total (Fig. 5). The 48-h interval was used for the first two passages of PD584 transformants and the PD689 control because these cultures initially showed slower growth than the cultures of PD689 transformants. By adopting this strategy, we aimed to select for cells that grew faster and reached a higher biomass yield on xylose than the template strains[20]. Twice during the experiment, glycerol stocks of the evolved cultures were prepared and individual clones were isolated. Firstly, when the OD$_{600}$ of a given culture reached the value of ≥ 3.5 within 24 h of growth, and secondly at the end of the experiment (after ~60–70 generations, Fig. 5). Cells were plated on M9 agar plates with 2 g L⁻¹ xylose and Km and 2–3 fastest growing clones from each cultivation were picked for further characterization. Growth of the selected clones on xylose was first tested in a 48-well plate format and nine fastest-growing candidates were verified in 24 h-long shake-flask cultures (Supplementary Fig. 4). The number of generations in the evolution experiment was calculated for each of the evolved strains using the Eq. (1)[4]:

$$\text{Number of generations} = \ln(\text{OD}_{600\text{final}}/\text{OD}_{600\text{initial}})/\ln(2) \qquad (1)$$

### Calculations of dry cell weight and growth parameters

Dry cell weight (DCW), maximal specific growth rate (μmax), lag phase (in h), biomass yield (Y$_{X/S}$), and biomass-specific substrate uptake rate (q$_s$) of characterized *P. putida* strains were determined as described in the Supplementary Method 7.

### ¹³C labeling experiments and analysis of metabolic fluxes

An initial pre-culture in LB medium was inoculated from a cryogenic glycerol stock of strain PD310 (Table 1) and incubated in a rotary shaker at 30 °C, 300 rpm and 5 cm amplitude. The second preculture was performed in 100 mL shake flask using 10 mL M9 medium with 5 g L⁻¹ xylose. The medium was inoculated with the first preculture to an OD$_{600}$ of 0.05 and incubated overnight under otherwise identical conditions as the LB preculture. Three 250 mL Erlenmeyer flasks containing 25 mL M9 medium were inoculated with the second overnight culture to an OD$_{600}$ of 0.025. The media contained 5 g L⁻¹ 1,2-¹³C xylose, i.e., xylose labeled with ¹³C isotopes at positions C1 and C2 (Sigma-Aldrich, 99% purity). Cultures were grown under agitation (300 rpm, amplitude: 5 cm) at 30 °C. Substrate uptake was monitored by HPLC-UV/RI analysis of the fermentation broth. Biomass concentrations were monitored using the cell growth quantifier (Scientific Bioprocessing) and determined manually by measuring the optical density at 600 nm with an Ultrospec 10 Cell Density Meter (GE Healthcare). The conversion factor of OD$_{600}$ to cell dry weight (CDW) in g/L was gravimetrically determined to 0.39.

During the exponential phase, OD and xylose concentrations were determined in 1 h intervals to allow accurate determination of growth and substrate uptake rates. At mid-exponential growth (OD$_{600}$ of 1.0–1.3), samples were taken to quantify the ¹³C isotope incorporation

into proteinogenic amino acids and free intracellular metabolites. A fast filtration method[73] was applied for intracellular metabolites to rapidly sample the biomass and quench the metabolism. Briefly, samples corresponding to 10 mg biomass were taken and the biomass was collected by vacuum filtration (Durapore, PVDF, 0.45 µm, 47 mm Sigma-Aldrich). After one washing step with 0.9% saline, the filter was placed (upside down) into a small Petri dish (5 cm diameter) filled with methanol (pre-cooled at −80 °C), incubated at −80 °C for 1 h. Filters were scraped and rinsed to ensure all cell material was in the liquid solvent, the solvent and filter were transferred to microfuge tubes and vigorously vortexed at −20 °C. The filter was removed, the extract centrifuged at 17,000 g at 4 °C for 5 min, and the supernatant removed and stored at −80 °C. Filter and solvent were transferred into a tube and vortexed at −20 °C. Extracts were dried in a lyophilizer and analyzed using capillary IC-MS analysis (Supplementary Method 8 and Supplementary Table 4, 5). The metabolites from upper glycolysis were important for a better resolution of fluxes of this part of the metabolism, especially cyclic fluxes. Capillary IC-MS data were corrected for the natural abundance of heavy isotopes using IsoCorr[74].

### Determination of $^{13}$C-labeling patterns in proteinogenic amino acids

A standard protocol was used as described in Schmitz et al.[75]. In brief, samples corresponding to 0.3 mg CDW were taken and centrifuged for 10 min at 4 °C at 17,000 g. The pellets were washed and resuspended in 5 N HCl, transferred to GC vials, and the samples were hydrolyzed at 105 °C for 6 h. Derivatization of the dried hydrolysate was done in a mix of 30 µL acetonitrile and 30 µL N-methyl-N-tert-butyldimethylsilyl-trifluoroacetamide (CS-Chromatographie Service GmbH) at 85 °C for 1 h. The derivatized samples were analyzed on a single-quadruple gas chromatography-mass spectrometry (GC-MS) system. Gas chromatography separation was performed using TRACE™ GC Ultra (Thermo Fisher Scientific, Waltham, MA, USA) equipped with an AS 3000 autosampler. The column used consisted of TraceGOLD TG-5SilMS fused silica (length, 30 m; inner diameter, 0.25 mm; film thickness, 0.25 µm). Helium was used as carrier gas at a constant gas flow rate of 1 mL min$^{-1}$ and a split ratio of 1:15. The injector temperature was set to 270 °C, and the column oven was heated according to a ramped program. The initial temperature of 140 °C was held for 1 min, the temperature was then increased at a rate of 10 °C/min to a final value of 310 °C, which was held time for 1 min. Mass spectrometry analysis utilized a Thermo Scientific ISQ single quadrupole mass spectrometer (Waltham, MA, USA). The transfer line and ion source temperatures were maintained at 280 °C, and ionization was achieved via electron impact (EI) ionization at 70 eV. Subsequent analysis of GC-MS raw data was conducted using Xcalibur software. Data were corrected for unlabeled biomass (introduced with the inoculum) and natural abundance of heavy isotopes using the software iMS2FLUX[76]. Metabolic flux analysis was performed using the Matlab-based tool INCA[77]. The model was constrained with mass isotopomer distribution data of intracellular metabolites and proteinogenic amino acids and the specific growth and xylose uptake rates. Confidence intervals were determined by parameter continuation using INCA's in-build function.

### Genome-scale metabolic model simulations

The most recent iJN1463 genome-wide metabolic reconstruction of *P. putida* KT2440[78] was downloaded from the BIGG database (http://bigg.ucsd.edu/). Two metabolites (xylose and xylulose) and four reactions – xylose transport to the periplasm and to cytosol (XYLtex, XYLt2pp respectively), xylose isomerase (XYLI1) and xylulokinase (XYLK) were added to the model. The glucose dehydrogenase reaction (GCD) was deleted from the model to prevent the oxidation of glucose to gluconate and 2-ketogluconate. To predict optimal flux distributions by flux balance analysis (FBA)[41] either COBRApy library or MATLAB COBRA toolbox[79] was used. Xylose uptake rate was fixed at

experimentally determined 1.45 mmol g$_{CDW}$$^{-1}$ h$^{-1}$. Model reactions of central carbon metabolism, TCA cycle, and CO$_2$ production were then constrained with upper and lower bounds as determined by metabolic flux analysis (Fig. 2). To get an upper and lower bound from MFA data, firstly, the standard error was calculated from two independent measurements and then the lower and upper bound on the fluxes were calculated as ± 1.96 × standard error (1.96 corresponds to the 97.5$^{th}$ percentile of a standard normal distribution). iJN1463 contains two malate dehydrogenase reactions (MDH and MDH2). The combined flux through these two reactions was set to the upper and lower bound calculated from MFA. The two models – one constrained with MFA data (MFA model) and another one constrained only with xylose uptake (FBA model) – were then compared (both modified models are available from GitHub). The glpk solver was used for all simulations. Maximizing the biomass formation rate was the objective function in all simulations.

### Whole-genome sequencing and proteomic analyses

Details on whole-genome sequencing of selected *P. putida* strains are provided in Supplementary Method 9. Details on proteome characterization of selected *P. putida* strains are provided in Supplementary Method 10 and Supplementary Table 6.

### Analytical methods

The optical density in cell cultures was recorded at 600 nm using UV/VIS spectrophotometer Genesys 5 (Spectronic). Analytes from cultures were collected by withdrawing 0.5 ml of culture medium. The sample was then centrifuged (20,000 g, 10 min). The supernatant was filtered through 4 mm / 0.45 µm LUT Syringe Filters (Labstore) and stored at −20 °C. Prior to the HPLC analysis, 50 mM H$_2$SO$_4$ in degassed miliQ water was added to the samples in a 1:1 ratio to stop any hydrolytic activity and to dilute the samples. High-performance liquid chromatography (HPLC) was used to quantify xylose and glucose. HPLC analysis was carried out using Agilent 1100 Series system (Agilent Technologies) equipped with a refractive index detector and Hi-Plex H, 7.7 × 300 mm, 8 µm HPLC column (Agilent Technologies). Analyses were performed using the following conditions: mobile phase 5 mM H$_2$SO$_4$, mobile phase flow 0.5 mL min$^{-1}$, injection volume 20 µL, column temperature 65 °C, RI detector temperature 55 °C. Xylose and glucose standards (Sigma-Aldrich) were used for the preparation of calibration curves. Xylose concentrations in labeling experiments were determined using a Beckman System Gold 126 Solvent Module equipped with a System Gold 166 UV-detector (Beckman Coulter) and a Smartline RI detector 2300 (Knauer). Analytes were separated on the organic resin column Metab AAC (Isera) eluted with 5 mM H$_2$SO$_4$ at an isocratic flow of 0.6 mL min$^{-1}$ at 40 °C for 40 min.

Glucose and xylose concentrations in culture supernatants were alternatively determined also by Glucose (GO) Assay Kit (Sigma-Aldrich, USA) and Xylose Assay Kit (Megazyme, Ireland), following the manufacturer´s instructions. Product concentrations were measured spectrophotometrically using Infinite M Plex reader (Tecan).

### Data and statistical analyses

The number of repeated experiments or biological replicates is specified in figure and table legends. The mean values and corresponding standard deviations are presented. When appropriate, data were treated with a two-tailed Student's *t*-test in Microsoft Office Excel 2013 (Microsoft) and confidence intervals were calculated for the given parameters to test a statistically significant difference in means between two experimental datasets.

### Reporting summary

Further information on research design is available in the Nature Portfolio Reporting Summary linked to this article.

## Data availability

All sequencing data and assembled whole-genome sequences were deposited under NCBI BioProject PRJNA914626. The detailed information can be found in Supplementary Table 1. The mass spectrometry proteomics data have been deposited to the ProteomeXchange Consortium via the PRIDE partner repository with the dataset identifier PXD047537. The raw gas chromatography-mass spectrometry and ion chromatography-mass spectrometry data used for the metabolic flux analysis in this study have been deposited in the Zenodo repository [https://zenodo.org/records/10732391]. Source data are provided with this paper.

## Code availability

Two modified genome-scale metabolic models of *P. putida* used in this study are available from GitHub [https://github.com/DalimilBujdos/P_putida_xylose_metabolism].

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

## Acknowledgements

We thank Dr. Adam Feist and Dr. Hyungyu Lim for valuable discussions on genomic data of engineered and evolved *P. putida* strains, and Dr. Ludmilla Aristilde for valuable discussion on flux analyses. This work was funded by Czech Science Foundation Project 22-12505 S and Grant Agency of Masaryk University GAMU Project MASH Junior 2022 (MUNI/J/0003/2021) granted to P.D. and Brno Ph.D. Talent granted to B.B. CIISB. This work was also supported by the project National Institute of Virology and Bacteriology (Programme EXCELES, ID Project No. LX22NPO5103), funded by the European Union - Next Generation EU. Instruct-CZ Centre of Instruct-ERIC EU consortium, funded by MEYS CR infrastructure project LM2023042, is gratefully acknowledged for the financial support of the measurements at the CEITEC Proteomics Core Facility. Computational resources were provided by the e-INFRA CZ project (ID:90254), supported by MEYS CR. We thank Dr Hendrik Ballersedt for support with GC-MS analysis and Tim Langhorst for esatablishing sampling procedures for intracellular metabolites.

## Author contributions

P.D., B.B., B.P., and B.E.E. contributed equally, they conceptualized the work, performed experiments, cured data, and drafted the manuscript. P.D., in addition, conceived the work, secured resources, and supervised B.B., B.P., D.B., and M.B. T.B. designed and performed whole genome sequencing and cured data. D.B. designed and performed genome-scale metabolic modeling and cured data. A. S.-P. contributed to the constructions of plasmids and strains. H.S. performed analyses (capillary IC-MS analysis), cured data. H.H. performed analyses (capillary IC-MS analysis), supervised H.S. and cured data. V. de L. contributed to the work conceptualization and supervised A.S.-P. L.M.B. contributed to the work conceptualization, read and approved the manuscript. M.B. contributed to the work conceptualization, plasmid and strain constructions, and cell cultures, and cured data. All authors read, edited and approved the final manuscript.

## Competing interests

The authors declare no competing interests.

## Additional information

**Supplementary information** The online version contains Supplementary Material available at https://doi.org/10.1038/s41467-024-46812-9.

