## [Peer Review File · Nature Communications]

Synthetically-primed adaptation of *Pseudomonas putida* to a non-native substrate D-xyloseReviewers' Comments:

Reviewer #1:

Remarks to the Author:

The paper from Dvořák and colleagues follows up on their previous work in xylose utilization in the robust soil bacterium, *Pseudomonas putida* KT2440. Overall, this is an elegant study and certainly worthy of publication in an outlet like Nature Communications. I have several comments that I hope will be of use to the authors, which are listed below in no particular order of importance. Please consider these comments as “minor” comments.

I do not really understand what “knowledge-primed” means. Do the authors mean to convey that they used knowledge gained to then take a next step (in this case in strain engineering and / or adaptive laboratory evolution)? If yes, that’s of course how science often (but not always) works, so maybe consider changing that? If it means something else, maybe the authors can explain better what they mean? This is obviously a minor point, but it is something that caught my attention as the title is funny to me.

This might be my misunderstanding / not relevant at all, but did the authors test xylose on an EM42 strain with the xylABE plasmid without the Δ gcd modification? Is that a reasonable control to include? I ask because the authors note that, on lines 5-7 of page 8 that PD310 is slower than previously developed strains. Certainly multiple studies have shown that production strains with Δ gcd grow poorer on glucose when using the KT2440 background (e.g., CW Johnson et al. Joule 2019) – are the authors’ results comparable to those studies where a lag is introduced on glucose by the Δ gcd modification? The authors compared growth of PD310 on glucose, which evidently is still good, so this makes me wonder further about how the Δ gcd and the +gcd strain compare on xylose.

Table 2 – do growth curves make more sense to show instead of a table? This is personal preference of course, but I typically prefer presenting these types of data in growth curves for a simpler visual comparison – the numbers can still be reported in a table in the SI or insets in a figure. This is just a suggestion and can obviously be ignored if the authors prefer the table format.

Page 9, lines 12-14 – wouldn’t it make more sense to compare xylose flux in a Δ gcd strain to glucose flux in a Δ gcd strain instead of in wild-type KT2440?

Page 16, lines 12-19 – the Ling et al. paper (ref. 5) included Δ hexR, for what it’s worth.

How do the best strains here handle mixed glucose and xylose?

Overall, this paper is extremely comprehensive and detailed, which I appreciate. At the end of the paper, I am left with the following questions:

- If I were starting from scratch and trying to engineer the best KT2440 (or EM42) strain to convert xylose to a product, what would I do with this information in hand?

- Would I use a strain from this study that has undergone ALE? If not, is there a way to make a “clean background” strain from these learnings?

For example, the ALE results show the replication of a large chunk of the genome, but the authors did not seemingly track down what in that replicated region (if one thing) is causing the observed phenotypic changes (or maybe they did this and I didn’t catch it). Reverse engineering of the responsible modifications into a clean background strain would have been really nice to see...it’s not essential, and it’s possible that the authors did something to this end that I’m not catching or that I’m misinterpreting. I also fully realize that this is a HUGE amount of work, so I’m not suggesting it must be done, but some explanation as to my questions above would be nice to include in the paper for

eventual readers of this work. I don't feel like the last paragraph before the methods section does this justice.

Reviewer #2:

Remarks to the Author:

The authors present adaptive laboratory evolution of *Pseudomonas putida* KT2440 for xylose utilization. They start with a previously engineered and evolved strain and present the continued improvement of it by a selection experiment or by using a selection experiment after rational engineering. *P. putida* has been previously adaptively evolved for xylose utilization also by other researchers. This created a complex starting situation for the manuscript. The noteworthy results of this study are the combinations of modifications that allowed improved growth of the *P. putida* strain KT2440 on xylose.

Please, find below my major and minor concerns on the manuscript detailed.

Major concerns

1. Authors present the strain optimization in an extended and combined Results and Discussion section. I would recommend dividing this into separate concise Results and Discussion -section as in the current form the results of this study, other works, and hypotheses are not clearly distinguishable for the reader. Furthermore, discussion on potential hypothesis is lengthy when not supported by the data.
2. The work integrates different means to elucidate the bottlenecks in the xylose utilization of the *P. putida* strain KT2440 (13C flux analysis, enzyme activity analyses, proteomics, whole-genome sequencing) but mechanisms hindering xylose utilization remain as hypotheses. On the other hand, the thorough work provides observations of significance to the particular field of *P. putida* metabolic engineering.
3. The authors have performed a selection experiment for two-weeks and estimate the number of generations passed during the two-weeks to be 60-70. The number of replicate lineages in the experiment remained unclear. Due to the short duration of the selection, the number of beneficial random mutations occurring and enriching in population is low and using the term adaptive laboratory evolution for the experiments appears misleading. The experiments appear as selection or competition experiments using parental populations that are variant libraries of unknown composition. Selection can be assumed to mostly act on phenotypes arising from the initial variants rather than phenotypes arising from random mutations having occurred during the experiment. Due to the initial populations being variant libraries, the whole genome sequencing has not been analyzed by variant calling against the parental genotypes. Thus, de novo mutations having occurred during the experiment cannot be concluded. In the analyses made, it was however, found that the intended talB-tktA cassette was not integrated in the strain PD584 tt L3, emphasizing that it remained unclear how the library composition was at the start of the selection experiment.

Minor concerns

1. Figure 2 and the 13C MFA result. It remains unclear how the PGI flux can be 89% as the incoming fluxes do not sum up to 89. Please, clarify.
2. Page 5, row 25: "EDEMP cycle" Please, clarify the abbreviation.
3. Page 13, row 5: "This difference in growth rate was consistently observed between FBA and MFA (constrained with MFA data) model results, even when relaxing (up to 5-fold, using the 5% increase step) the MFA flux constraints or xylose uptake rates (Supplementary Fig. S2)." The simulation approach and its aim remain unclear here. Please, clarify.

4. Page 15, row 10: "This could reflect a transient state induced by epigenetic regulation of these enzymes during growth on xylose⁴⁷." It remains unclear what is referred to with a transient state here. Please, clarify.

5. Page 19, row 8: "We combined this rational engineering step with ALE on xylose (Fig. 4c, Supplementary Fig. S5a)." It remains unclear here (found only in materials and methods) how the engineered variants were combined for ALE. Please, clarify.

Reviewer #3:

Remarks to the Author:

The paper describes the development of *Pseudomonas putida* strains based on the XI-pathway for more efficient assimilation of xylose. Dvorak and coworkers studied strains with recombinant XI-pathway, knockout of Gcd and other targets, and through ALE. Developed strain 310 from previous study was analysed with C13 metabolic flux analysis providing insight into metabolic responses to the xylose-perturbation. New strains were built with deletion of the transcriptional regulator hexR alone or with deletion of Gnd and subjected in parallel to new round of ALE. Characterisation of improved variants demonstrated differences in central carbon metabolism. The study is highly interesting and describe improved strains, and potential metabolic targets for improvement of xylose utilisation in *Pseudomonas*, which is highly interesting for biorefinery applications. The study would benefit from clearer description of novelty and main results, and more detailed/clearer directions for future engineering campaigns. See below specific comments

Major and minor comments:

Abstract: The abstract needs clearer description of the major finding(s). What is meant by that de-repressing glycolysis is needed to "unlock the route for xylose-derived carbon"? Also please describe with detail the unique insights mentioned on line 11. What engineering targets were identified in this study (not already known) to improve (or not improve) xylose utilisation?

Page 3. Line 5. "...non-model organisms." It is unclear what the authors mean by non-model organism. *Pseudomonas putida* could arguably be considered a model organism.

Title and introduction: Page 5. Line 15 "knowledge-primed engineering approach" (line 15), and the title is "Knowledge-Primed adaptation of *Pseudomonas*...". Please clarify the word "Knowledge-Primed" or consider changing the phrasing. It is unclear what exactly differs the approach from other engineering approaches that supposedly are not considered "knowledge-primed".

Page 5. Lines 15-Page 6. Line 3. This paragraph needs to be more clear to exactly what is done. Some parts is redundant. It is unclear what information presented in the study "is pivotal to further enhance xylose metabolism in *P. putida*". The section would also benefit from less colloquialism used in less informal contexts. For example, the following phrasing could be reconsidered: "explore how the metabolism of *P. putida*,... handles carbon flux from a non-native sugar substrate". Instead of "handles" perhaps "is adjusted", or "is perturbed", or "adapts to the perturbation of.." or similar is more correct?

Results and discussion. 2.1. Metabolic flux analysis: What key flux from this analysis in the EDMP include through the metabolite 6-phosphoglucose. Knockout of Pgi, Gnd and Edd individually were found to stop xylose assimilation. Impact of other elevated fluxes observed from MFA or were not studied. Interesting difference in malic enzyme between MFA and FBA could have been highlighted more. What is the effect of knocking out MaeB?

Table 2. Growth rate and lag phase is determined from cultivation in 48-well plates, and biomass yield in Erlenmeyer flasks. How is the oxygen transfer rate in these and does limitation in oxygen give an impact on the results? Oxygen is clearly important aspect of xylose metabolism. Was it considered throughout the experiments?

The XylABE were introduced in all strains in a multicopy plasmid carrying an antibiotic marker gene. What is the effect of plasmid copy number on the activity of XylABE in different strains? It is unclear to what extent the use of antibioticum in the media during ALE experiments influence the results. Is there any adaptation of the strains to the antibioticum, and does it give any impact on the xylose utilisation (e.g. through improved tolerance, or plasmid copy number)? Is there a correlation between

XylABE activity and concentration of the antibiotics? What ori is used and were mutations in regulatory system for plasmid replication observed?

Results and discussion (lines 3-15): Relative XylABE expression and growth on xylose. How different is EM42 from S12 and does it matter which strain background is used? It is mentioned that high activity of XylE is important, and this is made possible by synthetic RBS with a theoretical translation rate of 12,544 au. The relevance of the number is unclear since it is not compared to alternatives. Pls put it in context and mention what comparison is relevant, for example is the relative expression of XylAB important?

Table 1. Please indicate which strains have undergone ALE and from which starting point."

Fig 1. Would it be possible to illustrate duplication of PP2114-2219 in figure 1? What genes are present in this region?

Page 16. Line 19. Are there other own known phenotypes of the hexR deletion? Is it of importance for bioprocess applications?

Page 19. Lines 5-6. How many unique colonies were obtained from the random integration?

Page 20. Line 3-5. Can you prove that hexR deletion is needed to achieve the outcome?

Page 21. Lines 15-19. What is the effect of biofilm formation on ALE experiment. Was the mentioned mutation in LapA shifting ability to form biofilm? Was any biofilm observed throughout the ALE experiment and does it have any impact on the assimilation of xylose (e.g. from substrate or oxygen diffusion limitation).

Page 22. Lines 2-4. Are you sure the multiplication has a positive effect? Could it be masked by other changes contributing to the results? Is the duplication the cause for the shortened lag phase (EM42 vs PD310)? Does the fact that the multiplication was absent in PD689 speak in favour of a neutral impact under xylose utilisation conditions?

Page 24. Lines 11-17. Although the levels did not impacted the observed phenotype could have been caused by point mutations in key enzymes. Were any such mutations observed?

Page 24. Lines 18-24. The potential effect of 32bp XylA&RBS duplication is intriguing. Is the potential "leader peptide" observed in the proteomics data? Is it present as a "fusion" to XylA and is the increased level an effect of improved translation/folding rather than mRNA abundance? Would be interesting to see data on this.

Page 27. Lines 9-10. What is the importance of the transaldolase? Is this similar to other xylose-utilising species carrying the XI pathway?

Page 28. Lines 3-8. It is argued that the higher activity of the XylABE is caused by upregulation of the synthetic operon by TF-associated effect. Here it would be good to clarify the potential effects of plasmid copy number. Is the same effect seen in strains with chromosomally integrated XylABE operon?

Page 29. Line 6. What is the effect of introducing the rpoD mutation in the original strain EM42? Do you observe the same increase? Could be tested with the PEM7-GFP construct.

Page 29. Line 18. "Discomfort" is rather vague. There could be several reasons. Please clarify differences in production between strains. Which are the "reducing cofactor generating enzymes" in PD689 tt L1? Are they NADH or NADPH dependent and what is the redox cost of the pyocyanin pathway?

Figure 7. Visualisation of the pathway would be helpful.

Responses to the Reviewers' comments on ms. with ID NCOMMS-23-33047 entitled "Knowledge-primed adaptation of *Pseudomonas putida* to a non-native substrate D-xylose" submitted to Nature Communications and how they have been addressed in the revised version of the manuscript (Responses are below original comments in magenta and can be also found in separate Rebuttal Letter which is part of the revised submission)

We would like to express many thanks to the reviewers for their critical reading of the manuscript and very constructive comments, which helped us to improve its quality. We have made every possible effort, including additional experiments and changes in the manuscript texts and title, to respond to all the comments.

Reviewer #1 (Remarks to the Author):

The paper from Dvořák and colleagues follows up on their previous work in xylose utilization in the robust soil bacterium, *Pseudomonas putida* KT2440. Overall, this is an elegant study and certainly worthy of publication in an outlet like Nature Communications. I have several comments that I hope will be of use to the authors, which are listed below in no particular order of importance. Please consider these comments as “minor” comments.

We thank the reviewer for the overall positive evaluation of our manuscript.

1) I do not really understand what “knowledge-primed” means. Do the authors mean to convey that they used knowledge gained to then take the next step (in this case in strain engineering and/or adaptive laboratory evolution)? If yes, that’s of course how science often (but not always) works, so maybe consider changing that? If it means something else, maybe the authors can explain better what they mean? This is obviously a minor point, but it is something that caught my attention as the title is funny to me.

We agree with the reviewer that the term “Knowledge-primed adaptation...” may sound a bit clunky. To better articulate our conceptual and methodological framework, we have decided to modify the title to “Synthetically-primed adaptation of *Pseudomonas putida* to a non-native substrate D-xylose” We thus introduce in our study the term “synthetically-primed adaptation” (SPA). The correctness of the term was discussed with a native speaker and respected scientist.

SPA involves rationally introducing a precursor strain with specific heterologous genes and/or genomic modifications, predicted to elicit a desired metabolic phenotype if/when optimally integrated into the host biochemical network. Following this priming step, the strain is subjected to selective conditions, allowing it to undergo genetic fluctuations until an optimal parameter set is found through systematic exploration of the solution space in vivo within such an experimental system. This approach transcends traditional adaptive laboratory evolution by deliberately equipping the progenitor strain with predefined functionalities, steering the evolutionary process towards a targeted phenotype.

In brief, **SPA can be described as an evolutionary approach primed by rational metabolic engineering and introduced synthetic genetic constructs.** Such a definition is now used in the revised version of our manuscript, both in the Introduction (last paragraph) and in the extended concluding paragraph of the Results and Discussion section.

2) This might be my misunderstanding / not relevant at all, but did the authors test xylose on an EM42 strain with the xylABE plasmid without the Δ gcd modification? Is that a reasonable control to include? I ask because the authors note that, on lines 5-7 of page 8 that PD310 is slower than previously developed strains. Certainly multiple studies have shown that production strains with Δ gcd grow poorer on glucose when using the KT2440 background (e.g., CW Johnson et al. Joule 2019) – are the authors' results comparable to those studies where a lag is introduced on glucose by the Δ gcd modification? The authors compared growth of PD310 on glucose, which evidently is still good, so this makes me wonder further about how the Δ gcd and the +gcd strain compare on xylose.

Strain EM42 Δ gcd pSEVA2213_xylABE was constructed already in our previous study (Dvořák and de Lorenzo, 2018 DOI: 10.1016/j.ymben.2018.05.019) to demonstrate the difference between gcd+ and gcd- strains grown on xylose. We showed that the gcd+ strain is not of much use for the purpose of xylose utilisation, because its functional Gcd converts approx. 50% of used D-xylose to D-xylonate, which is (in contrast to D-gluconate) not utilised by *P. putida*. The initial growth of the gcd+ strain on xylose is comparable to the gcd- strain, but its final biomass is much lower due to the non-productive conversion of xylose to xylonate (please, see below graphs C, gcd- strain, and D, gcd+ strain, copied from Figure 4 in Dvořák and de Lorenzo, 2018). Therefore, we decided not to work with the gcd+ strain in the current study.

Regarding the growth of EM42 gcd- strain on glucose, indeed, we repeatedly observe a reduced growth rate of this strain when compared to EM42 gcd+. For instance, in our recent work by Bujdoš et al. 2023 (doi.org/10.1016/j.ymben.2022.10.011), we show that the gcd deletion reduced the growth rate of the strain by almost 40%. This result is comparable with studies using KT2440 strain.

Fig. 4. Growth of *P. putida* EM42 and its recombinants in minimal medium with 5 g L⁻¹ D-xylose. Experiments were carried out in shaken flasks at 30 °C and 170 rpm. (A) *P. putida* EM42, (B) *P. putida* EM42 Δ gcd pSEVA2213_xylAB, (C) *P. putida* EM42 Δ gcd pSEVA2213_xylABE, (D) *P. putida* EM42 pSEVA2213_xylABE. D-xylose, closed circles (●); D-xylonate, open circles (○); cell growth, closed squares (■). Data shown as mean \pm SD from three independent experiments.

3) Table 2 – do growth curves make more sense to show instead of a table? This is personal preference of course, but I typically prefer presenting these types of data in growth curves for a simpler visual comparison – the numbers can still be reported in a table in the SI or insets in a figure. This is just a suggestion and can obviously be ignored if the authors prefer the table format.

We prefer to show the summary of the growth parameters in Table 2. The comparison of all respective growth curves from plate reader cultures is actually shown later in Figure 4d and the shake-flask culture growth curves are summarized in the Supplementary Information (Supplementary Figures 1 and 6). Thus, all relevant growth curves are present in the manuscript (both in the main text and in SI) and, in addition, we provide the reader with the well-arranged summary of all relevant growth parameters in Table 2 positioned at the beginning of the Results and Discussion section.

4) Page 9, lines 12-14 – wouldn't it make more sense to compare xylose flux in a Δ gcd strain to glucose flux in a Δ gcd strain instead of in wild-type KT2440?

To the best of our knowledge, metabolic flux analysis of *P. putida* KT2440 Δ gcd grown on glucose has not yet been published. Hence, we cannot compare fluxes in our strain with the data from KT2440 Δ gcd. Nevertheless, it was not our intention in the part of the manuscript pointed out by the reviewer. On the contrary, we wanted to highlight the differences in fluxes between the Δ gcd strain grown on xylose (non-native substrate, uncommon distribution of fluxes) and wild-type strain grown on glucose (native substrate, "natural" distribution of fluxes).

5) Page 16, lines 12-19 – the Ling et al. paper (ref. 5) included Δ hexR, for what it's worth.

Good point! We know that the hexR deletion is mentioned in the paper by Ling et al. The hexR deletion and its positive effect on the growth of KT2440 Δ gcd strain on glucose was actually first reported by Bentley and co-workers (2020) (<https://doi.org/10.1016/j.ymben.2020.01.001>). Ling et al. included this mutation in their strain co-utilising glucose and xylose for muconic acid production due to its positive effect on glucose metabolism of the KT2440 Δ gcd strain reported previously by Bentley et al. However, Ling et al. did not discuss the effect of this deletion in the context of xylose metabolism. It was only Meijnen and colleagues (2012, DOI: 10.1074/jbc.M111.337501) who identified the HexR regulator as a potential target for improving xylose metabolism in *Pseudomonas*, but their study focused on *P. putida* S12 strain, not KT2440. We cite this work in our manuscript.

6) How do the best strains here handle mixed glucose and xylose?

Both best strains PD584 L3 and PD689 tt L1 handle the mixture of glucose and xylose well. As we mention in the main text of the manuscript (page 20 lines 15 and 16 in the original submission, the very last sentence in section 2.2 Knowledge-primed enhancement of xylose metabolism in *P. putida*) and show in the Supplementary information (Supplementary Figure S8 in the original submission, Supplementary Figure 7 in the revised manuscript) both strains are able to co-utilize glucose and xylose in the mixture (2 g/L of each sugar) in less than 20 h. Hence, they not only maintained the previously reported ability of the ancestral strain PD310 to co-utilize these sugars (Dvořák and de Lorenzo, 2018 DOI: 10.1016/j.ymben.2018.05.019) but they even outperformed this strain, which took 28 h to co-utilize the same amount of glucose and xylose (Dvořák and de Lorenzo, 2018).

Overall, this paper is extremely comprehensive and detailed, which I appreciate. At the end of the paper, I am left with the following questions:

7) If I were starting from scratch and trying to engineer the best KT2440 (or EM42) strain to convert xylose to a product, what would I do with this information in hand? Would I use a strain from this study that has undergone ALE? If not, is there a way to make a "clean background" strain from these learnings? For example, the ALE results show the replication of a large chunk of the genome, but the authors did not seemingly track down what in that replicated region (if one thing) is causing the observed phenotypic changes (or maybe they did this and I didn't catch it). Reverse engineering of the responsible modifications into a clean background strain would have been really nice to see...it's not essential, and it's possible that the authors did something to this end that I'm not catching or that I'm misinterpreting. I also fully realize that this is a HUGE amount of work, so I'm not suggesting it must be done, but some explanation as to my questions above would be nice to include in the paper for eventual readers of this work. I don't feel like the last paragraph before the methods

section does this justice.

Following the reviewer's recommendation and our own curiosity, we carried out the reverse engineering. We suspected that the transaldolase gene (PP_2168) present in the multiplied region was responsible for the improved growth on xylose. The gene was cloned into pSEVA438 plasmid and the pSEVA438_tal construct was inserted into the EM42 Δgcd pSEVA2213_xylABE strain without multiplication. In parallel, the same strain was transformed with the empty pSEVA438 plasmid. Comparison of the growth of these strains showed that the overexpression of the tal gene alone confers an improved phenotype (Fig. 7c in the revised manuscript), which in the case of PD310 was made possible by amplification of the entire 118 kb segment including tal gene.

To test for an additive effect of the duplication upstream of the xylA gene and tal overexpression, we prepared a new reverse-engineered strain PD855 by inserting the mutated pSEVA2213_xylABE plasmid with duplication and pSEVA438_tal into the freshly prepared strain EM42 $\Delta gcd \Delta hexR$ without multiplication in its chromosome (named PD580). We showed that the growth parameters of PD855 are close to those of the best strain obtained after ALE PD584 L3 (parameters and growth curves are shown in Tables 2, Fig. 4d, Supplementary Fig. 6 in the revised manuscript). Comparison of the growth of PD855 (lag phase 4.94 ± 0.39 h) with the same strain but with functional hexR (lag phase 20.80 ± 0.31 , Fig. 7d) also further highlighted the importance of the hexR deletion.

We now show that overexpression of the transaldolase gene was also important for the emergence of PD689 tt L1 on the PD689 genetic background (insertion of the pSEVA438_tal plasmid into PD689 significantly improved the growth of this strain, see Figure 7e in the revised manuscript). The increased expression of xylA in PD689 tt L1 was achieved by a more "brute-force" approach than in PD584 L3 - a missense mutation Ser552Pro in the sigma factor RpoD, which increased the affinity of the RNA-pol complex to the EM7 promoter of pSEVA2213_xylABE. We confirmed this by another additional experiment - transplantation of the Ser552Pro mutation into the PD580 strain, which then produced significantly more GFP from the pSEVA2213_EM7_gfp construct than the control PD580 strain without the Ser552Pro mutation in its rpoD gene (see Supplementary Fig. S11 in the revised manuscript).

In summary, we confirmed experimentally that the enhanced expression of transaldolase and xylose isomerase together with derepressed glycolysis were key events during the adaptation process. Moreover, we show that the two best *P. putida* strains, PD584 L3 and PD689 tt L1, obtained in this study occupy discrete local optima on the fitness landscape, achieved through different evolutionary paths. Our approach delivered strains with a doubled growth rate and substantially reduced lag phase on xylose, including a reverse-engineered strain PD855 with only two targeted deletions ($\Delta gcd \Delta hexR$), three exogenous genes (*xylABE*), and overexpressed endogenous transaldolase. These major findings are now highlighted in the Abstract, Results and discussion and the final concluding paragraph sections of the revised manuscript. We believe that these findings are important not only for future attempts to exploit semi-synthetic xylose metabolism in *P. putida* for biotechnological purposes, but also for the understanding of bacterial adaptation to new substrates.

Regarding the reviewer's question, we would recommend working with the PD584 L3 strain selected after ALE or with the PD855 strain reverse engineered on the "clean background".

Reviewer #2 (Remarks to the Author):

The authors present adaptive laboratory evolution of *Pseudomonas putida* KT2440 for xylose utilization. They start with a previously engineered and evolved strain and present the continued improvement of it by a selection experiment or by using a selection experiment after rational engineering. *P. putida* has been previously adaptively evolved for xylose utilization also by other researchers. This created a complex starting situation for the manuscript. The noteworthy results of this study are the combinations of modifications that allowed improved growth of the *P. putida* strain KT2440 on xylose.

Please, find below my major and minor concerns on the manuscript detailed.

Major concerns

1. Authors present the strain optimization in an extended and combined Results and Discussion section. I would recommend dividing this into separate concise Results and Discussion -section as in the current form the results of this study, other works, and hypotheses are not clearly distinguishable for the reader. Furthermore, discussion on potential hypothesis is lengthy when not supported by the data.

While the request of referee #2 to separate the results from the discussion is entirely legitimate, we believe that the narrative flow of the article is much more comprehensible if the experiments are reported and, if necessary, relevant comments are added on the spot. In our opinion, this is better than leaving it in a separate Discussion section later in the text, as the work described is quite complex and it might be distracting to leave the reader with some interpretation doubts until he or she reaches the final section of the article. Separation of the R&D section would also most likely lengthen the manuscript as description of some results would have to be (at least briefly) repeated also in the Discussion section.

Of course, it is also our primary interest to make the study as clear and understandable as possible. Hence, the entire text of the article has been edited for narrative coherence, and we believe that the revised version avoids some of the problems raised by the reviewer. Results of other works are always linked to the respective reference and in many cases also to the name of the first author (e.g., Meijnen et al., Elmore et al., or Bentley et al.). In contrast, our hypotheses are indicated by phrases “we hypothesize or we argue”, while our results and experimentally verified hypotheses are more often initiated with phrases such as “we determined, we prepared, we verified, we confirmed”. Moreover, we removed some speculative parts from our manuscript and conducted additional experiments that confirmed our key hypotheses (please, see the response to comment #2). The major findings are now highlighted in the Abstract and in the final concluding paragraph of the R&D section in the revised manuscript.

We, therefore, ask that the overall organization of the paper be left as it is. We believe that the changes address most of the reviewer's concerns regarding the separation of the R&D section. It is also worth mentioning that the joint R&D section is not uncommon in the Nature Communications journal. Here are several examples from the last journal issues:

<https://www.nature.com/articles/s41467-023-42208-3>
<https://www.nature.com/articles/s41467-023-42253-y>,
<https://www.nature.com/articles/s41467-022-29218-3>,
<https://www.nature.com/articles/s41467-022-35232-2>,
<https://www.nature.com/articles/s41467-022-30877-5>

2. The work integrates different means to elucidate the bottlenecks in the xylose utilization of the *P. putida* strain KT2440 (13C flux analysis, enzyme activity analyses, proteomics, whole-genome sequencing) but mechanisms hindering xylose utilization remain as hypotheses. On the other hand, the thorough work provides observations of significance to the particular field of *P. putida* metabolic engineering.

We take this comment to mean that in some cases we have not shown sufficient confidence in describing our results, especially the effect of targeted rational modifications and mutations that arise during the process of adaptive evolution, and have discussed our conclusions as hypotheses rather than as experimentally verified facts. As mentioned by reviewers, the study combines a wide range of methods and experiments by which we have verified our results and main conclusions. At the same time, and in light of other comments made by reviewer 2 and other reviewers, in recent months, we have added several additional experiments to the study to further verify some of our hypotheses.

Most importantly, we carried out reverse engineering, which showed that the overexpression of the *tal* gene alone confers an improved xylose utilization phenotype (please, see Fig. 7c and Supplementary Fig. 9 in the revised manuscript, and texts on page 22: lines 19-25, page 24: lines 12-19, page 27: lines 25-31), which in the case of PD310 was made possible by amplification of the entire 118 kb genomic segment including *tal* gene.

To test for an additive effect of the duplication upstream of the *xylA* gene and *tal* overexpression, we prepared a new reverse-engineered strain PD855 by inserting the mutated pSEVA2213_*xylABE* plasmid with duplication and a new plasmid with cloned transaldolase gene into the freshly prepared strain EM42 Δ *gcd* Δ *hexR* without multiplication in its chromosome. We showed that the growth parameters of PD855 are close to those of the best strain obtained after ALE - PD584 L3 (parameters and growth curves are shown in Tables 2, Fig. 4d, Supplementary Fig. 6, texts on page 27: lines 25-31 in the revised manuscript). Comparison of the growth of PD855 (lag phase 4.94 ± 0.39 h) with the same strain but with functional *hexR* (lag phase 20.80 ± 0.31 , Fig. 7d in the revised manuscript) also confirmed the importance of the *hexR* deletion.

In addition, we now show that enhanced expression of the transaldolase gene was also important for the emergence of PD689 tt L1 on the PD689 genetic background (overexpression of cloned transaldolase in PD689 significantly improved the growth of this strain, please, see Figure 7e in the revised manuscript). In another experiment, we confirmed that the Ser552Pro mutation found in the *rpoD* gene of the PD689 tt L1 strain is indeed responsible for enhanced expression of the whole synthetic *xylABE* operon (most probably through the increased affinity of RNA pol. complex to the EM7 promoter of pSEVA2213 plasmid, please, see discussion on page 30: lines 2-4 and Supplementary Fig. 11 in the revised manuscript).

In summary, we confirmed experimentally that the enhanced expression of transaldolase and xylose isomerase together with derepressed glycolysis were key events during the adaptation process. Moreover, we show that the two best *P. putida* strains, PD584 L3 and PD689 tt L1, obtained in this study occupy discrete local optima on the fitness landscape, achieved through different evolutionary paths. Our approach delivered strains with a doubled growth rate and substantially reduced lag phase on xylose, including a reverse-engineered strain PD855 with only two targeted deletions (Δ *gcd* Δ *hexR*) and three exogenous genes (*xylABE*), and overexpressed endogenous transaldolase. These major findings are now highlighted in the Abstract, Results and discussion section and its final concluding paragraph in the revised manuscript. We believe that these findings

are important not only for future attempts to exploit semi-synthetic xylose metabolism in *P. putida* for biotechnological purposes, but also for the understanding of bacterial adaptation to new substrates.

3. The authors have performed a selection experiment for two-weeks and estimate the number of generations passed during the two-weeks to be 60-70. The number of replicate lineages in the experiment remained unclear. Due to the short duration of the selection, the number of beneficial random mutations occurring and enriching in population is low and using the term adaptive laboratory evolution for the experiments appears misleading. The experiments appear as selection or competition experiments using parental populations that are variant libraries of unknown composition. Selection can be assumed to mostly act on phenotypes arising from the initial variants rather than phenotypes arising from random mutations having occurred during the experiment. Due to the initial populations being variant libraries, the whole genome sequencing has not been analyzed by variant calling against the parental genotypes. Thus, de novo mutations having occurred during the experiment cannot be concluded. In the analyses made, it was however, found that the intended talB-tktA cassette was not integrated in the strain PD584 tt L3, emphasizing that it remained unclear how the library composition was at the start of the selection experiment.

Thanks to reviewer #2 for the thought-provoking comment. Indeed, intrapopulation polymorphisms in the parental strains were not originally examined, so it was not clear whether de novo mutations or polymorphisms of the parental population fixed by selection were present in subsequent strains. Therefore, **we performed additional variant calling for the parental strains and comparative genomic analysis, which showed that all reported and discussed mutations arose during the adaptation phase, i.e. de novo.** The results obtained have been added as Supplementary File 4 and the text in Methods and Results and discussion sections has been modified accordingly (page 21: lines 20-24 in the revised ms). These results confirm that the duration of the experiment was sufficient to allow beneficial mutations to emerge and also confirm its designation as adaptive laboratory evolution (ALE).

There was indeed an initial period (4-5 days) of selection for the integration of *tktA-talB* and *tktA-talB-rpe-rpiA* cassettes (cassettes contain also streptomycin resistance gene) in the chromosome of the strains transformed with the pBAMD constructs (two antibiotics Km and Sm were present in the culture medium). ALE in the minimal medium with xylose and Km only then helped to balance the metabolism rewired by targeted deletions (Δ gcd, Δ hexR, Δ gnd) and introduced synthetic cassettes with exogenous genes (*xylABE*, *tktA-talB* or *tktA-talB-rpe-rpiA*). It yielded improved phenotypes from test (PD689 tt L1 with exogenous *tktA* and *talB* genes) as well as control (PD584 L3 without inserted *tktA-talB*) cultures. To clarify the ALE experiment, we added additional information to the Results and Discussion section in the form of Fig. 5 (please, see the revised version of the manuscript). The whole process is described in detail also in the Methods section.

Minor concerns

1. Figure 2 and the 13C MFA result. It remains unclear how the PGI flux can be 89% as the incoming fluxes do not sum up to 89. Please, clarify.

We would like to draw the reviewer's attention to the fluxes within PPP. F6P is formed in both the Tal (transaldolase, flux 44) and Tkt2 (transketolase, flux 42) reactions, and Fbp (flux 3) also contributes to the F6P pool. This gives a total of 89. It is possible that the reviewer was confused by the location of

the Tal reaction with its F6P product in another part of the figure. Plotting the complicated interrelated Tkt and Tal reactions in PPP is challenging. We have chosen this scheme variant, which we think is clear, but it should be noted that F6P and G3P are products of both Tal and Tkt reactions and are therefore mentioned in two places in the PPP diagram. We added this remark in the Figure 2 caption in the revised version of the manuscript.

2. Page 5, row 25: “EDEMP cycle” Please, clarify the abbreviation.

EDEMP cycle = cycle formed by the reactions of the pentose phosphate pathway, the Embden-Meyerhof-Parnas pathway, and the Entner-Doudoroff pathway. Following the journal’s instructions, we try to keep the Introduction concise but clear. Hence, we do not explain the abbreviation directly in the text, but we refer a reader to Figure 1, which is part of the Introduction and in which the cycle is depicted and described in detail.

3. Page 13, row 5: “This difference in growth rate was consistently observed between FBA and MFA (constrained with MFA data) model results, even when relaxing (up to 5-fold, using the 5% increase step) the MFA flux constraints or xylose uptake rates (Supplementary Fig. S2).” The simulation approach and its aim remain unclear here. Please, clarify.

We agree with the reviewer that the description of this particular simulation was not very clear and the simulation itself did not bring any new information. It was only an extension of the comparison of the two prepared models (MFA and FBA model) discussed in the same paragraph. Therefore, we removed this simulation from the manuscript.

4. Page 15, row 10: “This could reflect a transient state induced by epigenetic regulation of these enzymes during growth on xylose⁴⁷.” It remains unclear what is referred to with a transient state here. Please, clarify.

The transient state refers to phenotypic variation. We do not expect the observed differences in XylA and XylB enzyme activity in cells grown on glucose vs. xylose to be due to mutations generated during one cultivation. According to literature (please, see some relevant references below), it would be plausible to examine the possibility of epigenetic regulation to cause this phenotypic variation. Since experimental investigation of this issue would make for a new research project, we reserve this thought as engaging but we are not taking this particular idea (in contrast to other, key ideas) any further in this study to keep the manuscript concise. Perhaps the following sentence, which we use in the revised manuscript, could be better suitable:

“This difference may reflect a growth condition dependent epigenetic regulation of the respective genes.”

Beaulaurier, J., Schadt, E. E., & Fang, G. (2019). Deciphering bacterial epigenomes using modern sequencing technologies. *Nature Reviews Genetics*, 20(3), 157-172.

Breckell, G. L., & Silander, O. K. (2023). Growth condition-dependent differences in methylation imply transiently differentiated DNA methylation states in *Escherichia coli*. *G3*, 13(2), jkac310.

Bury-Moné, S., & Sclavi, B. (2017). Stochasticity of gene expression as a motor of epigenetics in bacteria: from individual to collective behaviors. *Research in Microbiology*, 168(6), 503-514.

5. Page 19, row 8: “We combined this rational engineering step with ALE on xylose (Fig. 4c,

Supplementary Fig. S5a).” It remains unclear here (found only in materials and methods) how the engineered variants were combined for ALE. Please, clarify.

To make this part of our manuscript easier to understand, we have moved Supplementary Figure 5, which describes the ALE process, from the Supplement to the main text and refer the reader to this figure (now Figure 5 in the revised manuscript).

Reviewer #3 (Remarks to the Author):

The paper describes the development of *Pseudomonas putida* strains based on the XI-pathway for more efficient assimilation of xylose. Dvorak and coworkers studied strains with recombinant XI-pathway, knockout of Gcd and other targets, and through ALE. Developed strain 310 from previous study was analysed with C13 metabolic flux analysis providing insight into metabolic responses to the xylose-perturbation. New strains were built with deletion of the transcriptional regulator hexR alone or with deletion of Gnd and subjected in parallel to new round of ALE. Characterisation of improved variants demonstrated differences in central carbon metabolism. The study is highly interesting and describe improved strains, and potential metabolic targets for improvement of xylose utilisation in *Pseudomonas*, which is highly interesting for biorefinery applications. The study would benefit from clearer description of novelty and main results, and more detailed/clearer directions for future engineering campaigns. See below specific comments

We thank the reviewer for the overall positive assessment of our manuscript. In the revised version, we focus on a clearer description of the novelty and main results, and we summarize the steps that should lead to improved xylose metabolism in engineering campaigns with *Pseudomonas* host.

Thanks to several additional experiments, we confirmed that the enhanced expression of transaldolase and xylose isomerase together with derepressed glycolysis were key events during the adaptation process to xylose. Moreover, we show that the two best *P. putida* strains, PD584 L3 and PD689 tt L1, obtained in this study occupy discrete local optima on the fitness landscape, achieved through different evolutionary paths. Our approach delivered strains with a doubled growth rate and substantially reduced lag phase on xylose, including a reverse-engineered strain PD855 with only two targeted deletions ($\Delta gcd \Delta hexR$) and three exogenous genes (*xylABE*), and overexpressed endogenous transaldolase. These major findings are now highlighted in the Abstract, Results and discussion section and its final concluding paragraph in the revised manuscript. We believe that these findings are important not only for future attempts to exploit semi-synthetic xylose metabolism in *P. putida* for biotechnological purposes, but also for the understanding of bacterial adaptation to new substrates.

Major and minor comments:

Abstract: The abstract needs clearer description of the major finding(s). What is meant by that de-repressing glycolysis is needed to “unlock the route for xylose-derived carbon”? Also please describe with detail the unique insights mentioned on line 11. What engineering targets were identified in this study (not already known) to improve (or not improve) xylose utilisation?

We modified the Abstract according to the reviewer’s comments. We just need to pinpoint that the new version is a compromise that takes into consideration also journal’s requirement to keep the Abstract concise (max. 150 words ideally).

We removed the colloquial expression “unlock the route”. Instead, we name specific targets of rational interventions such as HexR repressor or introduced transaldolase and transketolase genes. We newly mention reverse engineering that together with other analyses enabled detailed insight into the parallel paths of bacterial adaptation to the non-native carbon source, highlighting the particular modifications - enhanced expression of transaldolase and xylose isomerase together with derepressed glycolysis - as key events during the adaptation process.

Our results show that transaldolase alone is a key pentose phosphate enzyme whose activity is one of the major determinants of the xylose utilization rate. This will be instrumental for future engineering campaigns, considering that overexpression of both transketolase and transaldolase is common in the studies focused on pentose metabolism enhancement in bacteria (please, see some relevant references below). Besides the above findings, what makes our study attractive is the very comprehensive interdisciplinary approach we used to map the adaptation process primed by rational genetic interventions in two parental strains with different genetic backgrounds (PD584 and PD689) that led to the strains PD584 L3 and PD689 tt L1 with equally improved phenotypes. Within the scope allowed, we try to emphasize these points both in the Abstract and in other parts of the revised manuscript (including, for example, an expanded concluding paragraph of the R&D section).

Ling et al. Muconic acid production from glucose and xylose in *Pseudomonas putida* via evolution and metabolic engineering. *Nat Commun.* 2022 Aug 22;13(1):4925. doi: 10.1038/s41467-022-32296-y.

Zhu et al. The CRISPR/Cas9-facilitated multiplex pathway optimization (CFPO) technique and its application to improve the *Escherichia coli* xylose utilization pathway. *Metab Eng.* 2017 Sep;43(Pt A):37-45. doi: 10.1016/j.ymben.2017.08.003. Epub 2017 Aug 9.

Zhang et al. Metabolic Engineering of a Pentose Metabolism Pathway in Ethanologenic *Zymomonas mobilis*. *Science.* 1995 Jan 13;267(5195):240-3. doi: 10.1126/science.267.5195.240.

Page 3. Line 5. "...non-model organisms." It is unclear what the authors mean by non-model organism. *Pseudomonas putida* could arguably be considered a model organism.

We wanted to emphasize that the vast majority of similarly detailed studies focusing on adaptation to novel substrates have been carried out using the best-studied model organisms, namely *Escherichia coli*. However, we agree with the reviewer that *P. putida* KT2440 can be considered a model in certain areas of biotechnology (e.g. biodegradation or lignin valorization). Therefore, we removed this part of the sentence from the Abstract.

Title and introduction: Page 5. Line 15 "knowledge-primed engineering approach" (line 15), and the title is "Knowledge-Primed adaptation of *Pseudomonas*...". Please clarify the word "Knowledge-Primed" or consider changing the phrasing. It is unclear what exactly differs the approach from other engineering approaches that supposedly are not considered "knowledge-primed".

We agree with the reviewer that the term "Knowledge-primed adaptation..." may sound a bit clunky. To better articulate our conceptual and methodological framework, we have decided to modify the title to "Synthetically-primed adaptation of *Pseudomonas putida* to a non-native substrate D-xylose" We thus introduce in our study the term "synthetically-primed adaptation" (SPA). The correctness of the term was discussed with a native speaker and respected scientist.

SPA involves rationally introducing a precursor strain with specific heterologous genes and/or genomic modifications, predicted to elicit a desired metabolic phenotype if/when optimally integrated into the host biochemical network. Following this priming step, the strain is subjected to selective conditions, allowing it to undergo genetic fluctuations until an optimal parameter set is found through the systematic exploration of the solution space in vivo within such an experimental system. This approach transcends traditional adaptive laboratory evolution by deliberately equipping the progenitor strain with predefined functionalities, steering the evolutionary process towards a targeted phenotype.

In brief, **SPA can be described as an evolutionary approach primed by rational metabolic engineering and introduced synthetic genetic constructs** (in our case hexR or gnd deletions and synthetic xylABE operon or tktA-talB cassette). Such a definition is now used in the revised version of our manuscript, both in the Introduction (last paragraph) and in the extended concluding paragraph of the Results and Discussion section.

Page 5. Lines 15-Page 6. Line 3. This paragraph needs to be more clear to exactly what is done. Some parts is redundant. It is unclear what information presented in the study “is pivotal to further enhance xylose metabolism in *P. putida*”. The section would also benefit from less colloquialism used in less informal contexts. For example, the following phrasing could be reconsidered: “explore how the metabolism of *P. putida*,... handles carbon flux from a non-native sugar substrate”. Instead of “handles” perhaps “is adjusted”, or “is perturbed”, or “adapts to the perturbation of..” or similar is more correct?

We have modified the last paragraph of the introduction according to the reviewer's suggestions. We removed some "colloquialisms" ("handle", "boost", "draw the initial picture") and some redundant parts, and we were more specific in describing the results obtained and highlighting the main findings of the study. However, we also wanted to keep the paragraph and the whole introduction concise (following the journal's instructions) and appealing to the reader (not listing too many details that are later revealed in the Results and Discussion section). Therefore, the revised version is a compromise between the reviewer's suggestions and the journal's criteria.

Results and discussion. 2.1. Metabolic flux analysis: What key flux from this analysis in the EDMP include through the metabolite 6-phosphoglucose. Knockout of Pgi, Gnd and Edd individually were found to stop xylose assimilation. Impact of other elevated fluxes observed from MFA or were not studied. Interesting difference in malic enzyme between MFA and FBA could have been highlighted more. What is the effect of knocking out MaeB?

Indeed, there were many differences between the fluxes calculated in MFA and FBA. It was not within our time and budget to verify the significance of all reactions that showed differences in flux levels between MFA and FBA and to perform targeted deletions of the corresponding genes. Therefore, we focused on a few key reaction nodes in the EDMP cycle that represent the upper metabolism of xylose and were expected to play a critical role in its processing by the cell. The distribution of fluxes in the EDMP cycle influences the flux distribution in the TCA cycle. The difference in flux through the MaeB reaction between MFA and FBA is indeed significant. High MaeB activity and contribution to the pyruvate pool is not unusual - it has been previously reported for *P. putida* grown on glucose (doi: 10.1074/jbc.M115.687749, doi: 10.1016/j.ymben.2019.01.008.). The effect of deleting the MaeB reaction could not be simulated because the genome-scale metabolic model without this reaction does not give a result in FBA. We did not delete the maeB gene because we suspected that such a deletion would lead to a bottleneck in the TCA cycle. The high flux from this reaction would have to be redirected to the Mdh malate dehydrogenase reaction, which already had a high flux in the PD310 strain. As we decided not to perform this deletion, we have not devoted further space to discussing the difference between MFA and FBA fluxes in this reaction. Such a discussion would be rather speculative and would further lengthen the manuscript, which we prefer to keep concise (as required by the journal).

Table 2. Growth rate and lag phase is determined from cultivation in 48-well plates, and biomass

yield in Erlenmeyer flasks. How is the oxygen transfer rate in these and does limitation in oxygen give an impact on the results? Oxygen is clearly important aspect of xylose metabolism. Was it considered throughout the experiments?

This is clearly an important point. *P. putida* is an obligate aerobe. Therefore, we chose a 48-well plate setup with intense continuous orbital shaking (245 rpm, 2.5 um amplitude) and aeration-enabling lid rather than a 96-well plate. Based on our experience, *P. putida* cultures grown in 48-well microplates show very similar growth rates to those grown in Erlenmeyer flasks under the conditions described in the Materials and methods section of our manuscript (we refer the reviewer to the Source Data file provided, in which the growth parameters of all strains from Table 2 are calculated for both plate and flask setups). The values for growth rate and lag phase in Table 2 were calculated from the data measured in the plate because we consider the parameters calculated from the larger amount of data collected during such automated OD measurements to be more accurate.

The XylABE were introduced in all strains in a multicopy plasmid carrying an antibiotic marker gene. What is the effect of plasmid copy number on the activity of XylABE in different strains? **It is unclear to what extent the use of antibioticum in the media during ALE experiments influence the results. Is there any adaptation of the strains to the antibioticum, and does it give any impact on the xylose utilisation (e.g. through improved tolerance, or plasmid copy number)? Is there a correlation between XylABE activity and concentration of the antibiotics?** What ori is used and were mutations in regulatory system for plasmid replication observed?

Good point. All strains used in our study contained the low copy number pSEVA2213 plasmid with RK2 ori and subcloned synthetic xylABE operon. We did not detect any mutation in the plasmid in any of the strains sequenced, with the exception of PD584 L3, which contained a 32 bp duplication upstream of the xylA gene (this mutation is discussed in detail in our study - see pages 25-27). Therefore, there is no evidence of a change in the copy number of the plasmid during the engineering and evolutionary process and selection on the antibiotic kanamycin. To be sure, we performed an additional experiment in which we isolated plasmid DNA from PD584, PD689 (parental strains before ALE) and PD584 L3 and PD689 tt L1 (strains selected after ALE) cells collected at exponential growth phase (OD600=0.5, 5 mL of culture) and quantified the DNA using Qubit fluorometric quantification (see the Graph and Figure below). We also compared the percentage of Illumina reads covering the plasmid in the three most extensively sequenced strains (see Table below). The results show no significant difference in plasmid DNA concentrations and reads. We conclude that the improved xylose utilization by the strains after ALE is not related to changes in plasmid copy number.

Table: Percentage of total reads (%) mapped to plasmids in selected *P. putida* strains determined by Illumina sequencing.

strain		Illumina seq.
PD584	reads	3.14%
PD689 tt L1	reads	2.42%
PD584 L3	reads	3.39%

Graph: The pSEVA2213_xylABE plasmid DNA concentration determined by the Qubit fluorometric quantification in the selected *P. putida* strains. The columns show the mean \pm standard deviation calculated from six biological replicates (individual data points are shown as blank circles). No statistically significant difference between the two means was identified (calculated using a two-tailed Student *t* test, $p \geq 0.05$). Source data are provided in the revised Source Data file (Review experiment 1).

Figure: Agarose gel with undigested pSEVA2213_xylABE plasmids isolated from the selected *P. putida* strains. The individual samples of the plasmid isolated from a particular strain and used for the fluorometric quantification (the Graph above) were pooled, concentrated using a vacuum evaporator (DNA120 SpeedVac, Savant®) and loaded on 0.8 % gel.

In addition, we tested the growth of four strains - parental strains PD584 and PD689 used as templates for ALE and evolved strains PD584 L3 and PD689 tt L1 - in different concentrations of kanamycin up to 1,000 $\mu\text{g}/\text{mL}$ (which is 20 times higher concentration than we use routinely in our cultures of *P. putida* strains with pSEVA2213 plasmid). The cultures were conducted on glucose as all four strains grow on this substrate similarly and the final OD after 24 h would not vary as in the case of xylose. We observed no difference in culture ODs among these strains even at the highest antibiotic concentration (see the table below). Hence, we believe that there is no difference in sensitivity to kanamycin between evolved strains and their parents and it should have no effect on xylose utilization phenotype at an experimental kanamycin concentration of 50 $\mu\text{g}/\text{mL}$.

Table: Growth of selected *P. putida* strains in minimal salts 2.5 mL M9 medium with 2 g/L glucose and different concentrations of kanamycin (cultures were conducted in 15 mL plastic tubes). Values of OD600 measured after 24 h of growth are means from two biological replicates.

Km concentration $\mu\text{g/mL}$	Optical density at 600 nm			
	PD584	PD689	PD584 L3	PD689 tt L1
50	1.49	1.53	1.50	1.48
250	1.52	1.50	1.52	1.43
500	1.50	1.57	1.58	1.50
750	1.48	1.50	1.54	1.47
1000	1.54	1.49	1.56	1.49
ctrl on M9+glucose a/b-	1.47	1.48	1.35	1.34

Results and discussion (lines 3-15): Relative XylABE expression and growth on xylose. How different is EM42 from S12 and does it matter which strain background is used? It is mentioned that high activity of XylE is important, and this is made possible by synthetic RBS with a theoretical translation rate of 12,544 au. The relevance of the number is unclear since it is not compared to alternatives. Pls put it in context and mention what comparison is relevant, for example is the relative expression of XylAB important?

Yes, the genetic background of the strains used is important. S12 and KT2440 (or EM42 as its genome-streamlined derivative Martínez-García et al. 2014 DOI: 10.1186/s12934-014-0159-3) are different strains. We aligned the two genomic sequences of strains S12 and EM42 using local alignment with orthoANI (DOI: 10.1007/s10482-017-0844-4, which is a standard method for such purposes). Due to the different lengths of the two sequences, only 85% of the genome was aligned, and in this aligned region 97.5% similarity was identified. This may not seem like a big difference, but as we show in our study, even a single nucleotide change or a mobile element-driven duplication of a genomic segment (a change that is often missed during genome assembly because it is only manifested in sequencing by increased coverage of the duplicated region) can have a significant effect on phenotype, including sugar utilization capacity. S12 and KT2440 (or EM42) share some characteristics - they are robust strains resistant to oxidative stress, they do not naturally use xylose. However, their genetic backgrounds are different and so can their ability to adapt to the new substrate or metabolic perturbations.

The theoretical translation rate for the xylE gene with the synthetic RBS used was calculated using the RBS Calculator (DOI: 10.1038/nbt.1568). This strong RBS from the pET expression vector was chosen to provide more than an order of magnitude higher theoretical translation rate (12,544 a.s.) than the RBSs of the preceding xylA and xylB genes (392 a.s. and 546 a.s., respectively), whose strengths corresponded to the native RBS preceding xylA and xylB genes in *E. coli*. To avoid a long description and to keep the manuscript concise, we decided to omit a specific number, which is only a theoretical estimate anyway. The corresponding sentence was changed to:

"In strain PD310, we expected sufficient expression of the xylE gene as it is controlled by the strong constitutive PEM7 promoter and a strong synthetic ribosome binding site (RBS)".

Table 1. Please indicate which strains have undergone ALE and from which starting point."

Fig 1. Would it be possible to illustrate duplication of PP2114-2219 in figure 1? What genes are present in this region?

This seems to be a misunderstanding. The information requested by the reviewer is already included in Table 1 (we describe in the table that strains PD584 L3, 584 tt L3 and PD689 tt L1 are derivatives of the respective parental strains PD584 and PD689 and were selected after the ALE process).

We prefer not to illustrate the multiplication of the PP_2114-PP_2219 region in Figure 1 to keep the timeline of our work and findings clear. This figure shows the upper sugar metabolism of *Pseudomonas putida* EM42 PD310 with the xylose isomerase pathway inserted and the corresponding pSEVA2213_xylABE plasmid construct used to implant the pathway. This figure does not illustrate the full genetic background of this strain. The multiplication was revealed later in our study and is discussed in the Results and Discussion section. In the relevant part of the Results and Discussion section, reference is also made to Supplementary File 5, which lists all the annotated genes present in the multiplied genomic segment.

Page 16. Line 19. Are there other known phenotypes of the hexR deletion? Is it of importance for bioprocess applications?

HexR is a repressor of some of the genes of the Entner-Doudorof and pentose phosphate pathway enzymes (*gap-1*, *edd*, *glk*, *zwf-1*, *pgl*, *eda*), *gltR-2* regulator and a hypothetical protein PP_4488. In general, HexR is part of the carbon catabolite repression system in *P. putida*, which prefers organic acids above glucose (DOI: 10.1111/1751-7915.13263). Deletion of HexR was reported in several studies on *P. putida* KT2440 as it was shown to improve growth and production of muconic acid (DOI: 10.1016/j.ymben.2020.01.001) or PHBA (DOI: 10.3389/fbioe.2016.00090) from glucose. In a couple of studies, *hexR* deletion was also correlated with increased resilience to oxidative stress (DOI: 10.1007/s12275-010-0075-0, DOI: 10.1099/mic.0.2008/020362-0). This was explained by overexpression of *zwf-1*, which leads to an increased pool of NADPH. This is of potential importance for bioprocess applications. However, as we need to keep our manuscript and reference list concise following the journal's instructions, we decided not to add these additional references to our study.

Page 19. Lines 5-6. How many unique colonies were obtained from the random integration

The transformation of PD584 and PD689 strains with the pBAMD_tktA-talB or pBAMD_tktA-talB-rpe-rpiA constructs was immediately followed by an adaptive laboratory evolution experiment and isolation of individual clones from cultures showing improved growth on xylose (the process is described in Fig. 5 in the revised manuscript). We plated a small part of the cells immediately after electroporation of pBAMD constructs into the PD584 and PD689 strains to check transformation efficiency (which ranged from 7×10^4 to 1×10^6 colonies / 1 μ g plasmid DNA) but we did not check the "uniqueness" of individual clones by checking the integration of *tktA-talB* or *tktA-talB-rpe-rpiA* expression cassettes into their chromosomes (which can be done either by arbitrary PCR or whole-genome sequencing, DOI: 10.1007/978-1-61779-412-4_16). Random integration of genes into *P. putida* chromosome via pBAMD tool is reliable, was well documented in previous studies (e.g., DOI: 10.3389/fbioe.2014.00046 or DOI: 10.1016/j.ymben.2022.10.011), and we have a good experience with this protocol. We did not expect a bottleneck in the integration step. The fact that the integration of *tktA-talB* or *tktA-tal-rpe-rpiA* cassette indeed occurred is documented by the isolation and sequencing of two strains PD689 tt L1 (with the *tktA-talB* cassette integrated into the PP_1181 gene encoding a two-component system response regulator) and PD689 ttr L2 (with the *tktA-tal-rpe-rpiA* cassette integrated into the *rapA* gene PP_1145) from the end of the evolutionary experiment. Out of these two, only PD689 tt L1 strain is mentioned in the submitted version of our manuscript because the growth of the PD689 ttr L2 strain on xylose was not stable. We attributed this unstable phenotype of PD584 ttr L2 to the interruption of the *rapA* gene. RapA is a transcription regulator that stimulates

RNA polymerase recycling and the disruption of the corresponding gene may thus cause discrepancies in global gene expression pattern (DOI: 10.1016/j.bbagr.2011.03.003).

Page 20. Line 3-5. Can you prove that hexR deletion is needed to achieve the outcome?

In our study, we show the positive effect of the hexR deletion on xylose in several independent experiments. When we compare PD310 (template strain without hexR deletion) and PD584 (derived strain with hexR deletion) and when we compare PD506 (template strain without hexR deletion) and PD689 (derived strain with hexR deletion). Both strains PD584 and PD689 grow significantly better (have a much shorter lag phase) than their templates (please, see Table 2). In the case of PD584, we also see increased activities of the EDEMP cycle enzymes Zwf and Edd-Eda, whose expression was derepressed upon hexR deletion (Figure 4b). These enzymes, when freed from HexR repression, can provide a pull for the carbon from xylose into the EDEMP cycle and accelerate growth. We also show a positive effect of hexR deletion on the growth on D-fructose, which is metabolized through the EDEMP cycle in a similar way to xylose (Supplementary Fig. 2 in the revised version). **A recent comparison of the growth of reversed-engineered strain PD855 (lag phase 4.94 ± 0.39 h) with the same strain but with functional hexR (lag phase 20.80 ± 0.31 , Fig. 7d in the revised manuscript) further highlighted the importance of the hexR deletion.**

These results are consistent with the observations of other authors that we cite in our manuscript. Meijnen and colleagues (2012, DOI: 10.1074/jbc.M111.337501) demonstrated with a transcriptomic analysis down-regulation of *hexR* in their evolved *P. putida* S12 strain with enhanced xylose utilization phenotype. Bentley and co-workers (2020, DOI: 10.1016/j.ymben.2020.01.001) and later Ling and colleagues (2022, DOI: 10.1038/s41467-022-32296-y) deleted hexR to improve the growth of their *P. putida* KT2440 Δ *gcd* mutant used for the production of muconic acid from glucose or a mixture of glucose and xylose, respectively. They demonstrated the beneficial effect of hexR deletion on glucose metabolism, but did not investigate its effect on xylose metabolism in KT2440. We mention the importance of the hexR deletion during the adaptation process on xylose in the Abstract and all following sections of our manuscript (Introduction, Results and discussion section and its final concluding paragraph).

However, as our data show that the hexR deletion was only one of the key events during the adaptation process (together with increased transaldolase and xylose isomerase activities), we decided to modify the claim referred to by the reviewer (page 19, lines 14-17 in the revised ms):

“A 3-week control ALE of *P. putida* PD310 did not result in enhanced growth of this strain on xylose (Supplementary Fig. S5). This suggests that the hexR deletion in PD584 and PD689 is an important factor in the enhanced carbon passage through the native glycolysis reactions (namely, the Zwf-Pgl-Edd-Eda part of the EDEMP cycle) and fast evolution of these strains.”

Page 21. Lines 15-19. What is the effect of biofilm formation on ALE experiment. Was the mentioned mutation in LapA shifting ability to form biofilm? Was any biofilm observed throughout the ALE experiment and does it have any impact on the assimilation of xylose (e.g. from substrate or oxygen diffusion limitation).

We thank the reviewer for this interesting thought. We did not observe any significant production of biofilm visible by the naked eye during the ALE experiment. Nevertheless, following the reviewer's suggestion, we decided to perform an additional experiment to check the biofilm formation capacity

of strains PD584 (template strain for ALE and control) and PD584 L3 and PD689 tt L1 (strains selected after ALE with numerous mutations in the *lapA* gene). We used a standard microtiter plate biofilm assay with crystal violet dye (DOI: 10.3791/2437), which is now described in the revised Source data file (Excel sheet Review experiment 2) together with the results of the experiment (see also the graph below). We did not observe a significant difference in the biofilm formation capacity of the three compared strains. Hence, we decided to soften our claim in the respective section of the manuscript (page 22, lines 3-5) to:

“Mutations in biofilm genes can emerge in response to stress or selection of planktonic cells during multiple transfers in ALE experiments, as was observed previously.”

This claim is linked to two relevant references:

Bentley et al. Engineering glucose metabolism for enhanced muconic acid production in *Pseudomonas putida* KT2440. *Metab Eng.* 2020 May;59:64-75. doi: 10.1016/j.ymben.2020.01.001.

Chu et al. Self-induced mechanical stress can trigger biofilm formation in uropathogenic *Escherichia coli*. *Nat Commun.* 2018 Oct 5;9(1):4087. doi: 10.1038/s41467-018-06552-z.

Graph: Biofilm formation of three *P. putida* strains grown on 2 g/L xylose in a 96 microtitre plate, assessed by crystal violet staining. Columns represent means \pm standard deviations calculated from five biological replicates (each of two technical replicates).

Page 22. Lines 2-4. Are you sure the multiplication has a positive effect? Could it be masked by other changes contributing to the results? Is the duplication the cause for the shortened lag phase (EM42 vs PD310)? Does the fact that the multiplication was absent in PD689 speak in favour of a neutral impact under xylose utilisation conditions?

Yes, we are sure the multiplication has a positive effect on the xylose utilization phenotype of our strains. Following the recommendation of reviewer #1 and our own curiosity, we carried out the reverse engineering. We suspected that the transaldolase gene (PP_2168) present in the multiplied region was responsible for the improved growth on xylose. The gene was cloned into pSEVA438 plasmid and the pSEVA438_tal construct was inserted into the EM42 Δ gcd pSEVA2213_xylABE strain without multiplication. In parallel, the same strain was transformed with the empty pSEVA438 plasmid. Comparison of the growth of these strains showed that the overexpression of the tal gene

alone confers an improved phenotype (Fig. 7c in the revised manuscript), which in the case of PD310 was made possible by amplification of the entire 118 kb segment including tal gene (the emergence of the multiplication in several freshly prepared *P. putida* EM42 Δ gcd pSEVA2213_xylABE clones was reproduced in another additional experiment described in Supplementary Fig. 9, Table S5 and page 24, lines 12-19 of the revised manuscript).

To test for an additive effect of the duplication upstream of the xylA gene and tal overexpression, we prepared a new reverse-engineered strain PD855 by inserting the mutated pSEVA2213_xylABE plasmid with duplication and pSEVA438_tal into the freshly prepared strain EM42 Δ gcd Δ hexR without multiplication in its chromosome. We showed that the growth parameters of PD855 are close to those of the best strain obtained after ALE PD584 L3 (parameters and growth curves are shown in Tables 2, Fig. 4d, Supplementary Fig. 6 in the revised manuscript).

We show that increased expression of the transaldolase gene was also important for the emergence of PD689 tt L1 on the PD689 genetic background (insertion of the pSEVA438_tal plasmid into PD689 significantly improved the growth of this strain, see Figure 7e in the revised manuscript).

Page 24. Lines 11-17. Although the levels did were not impacted the observed phenotype could have been caused by point mutations in key enzymes. Were any such mutations observed?

No, all mutations in coding sequences are listed in Supplementary File 4. No mutation, which could explain improvement was identified.

Page 24. Lines 18-24. The potential effect of 32bp Xyla&RBS duplication is intriguing. Is the potential “leader peptide” observed in the proteomics data? Is it present as a “fusion” to XylA and is the increased level an effect of improved translation/folding rather than mRNA abundance? Would be interesting to see data on this.

The theoretical maximum length of the expected leader peptide that could be formed by the duplication event is 32 amino acids (please, see figure above), then translation would be stopped by the premature STOP codon. Leader peptide is probably not present as a fusion with XylA, because it is out of the frame of the xylA gene. Several variants of the expected leader peptide (ending with K or R, due to trypsin digestion prior to proteomic analysis) could theoretically be detected by proteomics:

MQAYSSKRNPCKPILTSSIAFVMKAQNPQTR

MQAYSSKRNPCKPILTSSIAFVMK

MQAYSSKRNPCK

MQAYSSKR

MQAYSSK

None of these peptides was detected in our original proteomic data neither during the new targeted proteomic analysis conducted in reaction to the reviewer's comment (all proteomics data have been deposited to the ProteomeXchange Consortium via the PRIDE partner repository with the dataset identifier PXD047537 and are accessible with the following login details mentioned in the revised version of the manuscript: Username: reviewer_pxd047537@ebi.ac.uk, Password: Q59EU08).

Thus, the effect of the duplication remains unclear and may be the subject of a separate future study in our laboratory. It is possible that a mechanism similar to a translational coupler, such as upstream translation-dependent de novo initiation, may have ensured better translation efficiency of xylA alone by "detangling" the secondary structure of the 5' UTR. Such mechanisms are described for example here:

Huber, M., Faure, G., Laass, S., Kolbe, E., Seitz, K., Wehrheim, C., ... & Soppa, J. (2019). Translational coupling via termination-reinitiation in archaea and bacteria. *Nature communications*, 10(1), 4006.

Flower, A. M., & McHenry, C. S. (1990). The gamma subunit of DNA polymerase III holoenzyme of *Escherichia coli* is produced by ribosomal frameshifting. *Proceedings of the National Academy of Sciences*, 87(10), 3713-3717.

As we could not confirm the presence of the leader peptide, we decided to soften our claim in the manuscript. The revised version (which can be found on page 25, lines 21-25) reads as follows

"It is possible that the effect of the duplication lies in the emergence of a mechanism similar to translational coupler – that is, a region downstream of the promoter that encodes a short leading peptide stabilizing translation of the downstream gene. However, no such peptide was detected in our proteomic data, so the molecular effect of the duplication remains to be elucidated."

Nevertheless, since the duplication is the only mutation detected on the pSEVA2213_xylABE plasmid isolated from the PD584 L3 strain, and we have shown that the mutant plasmid allows faster growth on xylose (Supplementary Fig. 10), we can say with high confidence that the duplication is responsible for the observed higher XylA abundance (Fig. 6b, Supplementary file 2) and activity (Fig. 7g) and improved growth on xylose of PD584 L3 strain.

Page 27. Lines 9-10. What is the importance of the transaldolase? Is this similar to other xylose-utilising species carrying the XI pathway?

After additional experiments described in one of the previous responses, it is clear that transaldolase is a key determinant of the xylose utilization phenotype. As we discuss in the Results and discussion section of our manuscript (page 28, lines 14-25), in contrast to PD310, PD584, and PD584 L3, which benefited from several copies of the native *tal* gene in the multiplied chromosome segment, the PD689 tt L1 strain enhanced the efficiency of the non-oxidative pentose phosphate pathway (PPP) in by the enzymes transplanted from *E. coli*. This is further supported by the measurements of transaldolase activity in six *P. putida* strains discussed in our study (Fig. 7g). Utilization of the exogenous TalB in PD689 tt L1 appears to be an adaptive response, necessitated by the absence of the genomic multiplication and the Gnd enzyme in its parental strain PD689. In mutants with *gnd* removed, the absence of 6-phosphogluconate dehydrogenase activity had to be compensated. The implanted TalB could provide the necessary pull effect to promote a flow of carbon through the preceding PPP reactions starting with xylulose 5-phosphate. Moreover, the adoption of two

exogenous genes could be a more economical solution in terms of cellular resource use than the amplification of a large genomic region.

PPP and its enzymes including transaldolase are fundamental components of cellular metabolism in all three domains of life (doi: 10.1111/brv.12140). We expect transaldolase to play an important role in other xylose-utilizing species, and there are a few such reports in literature:

Gu et al. Improvement of xylose utilization in *Clostridium acetobutylicum* via expression of the *talA* gene encoding transaldolase from *Escherichia coli*. *J Biotechnol.* 2009 Sep 25;143(4):284-7. doi: 10.1016/j.jbiotec.2009.08.009.

Walfridsson et al. Xylose-metabolizing *Saccharomyces cerevisiae* strains overexpressing the *TKL1* and *TAL1* genes encoding the pentose phosphate pathway enzymes transketolase and transaldolase. *Appl Environ Microbiol.* 1995 Dec;61(12):4184-90. doi: 10.1128/aem.61.12.4184-4190.1995.

Page 28. Lines 3-8. It is argued that the higher activity of the XylABE is caused by upregulation of the synthetic operon by TF-associated effect. Here it would be good to clarify the potential effects of plasmid copy number. Is the same effect seen in strains with chromosomally integrated XylABE operon?

We have already refuted the potential effect of changing the plasmid copy number of plasmid pSEVA2213_xylABE in our answer to the previous reviewer's question. In another additional experiment, we confirmed that the Ser552Pro mutation found in the *rpoD* sigma factor gene of the PD689 tt L1 strain is indeed responsible for enhanced expression of the whole synthetic xylABE operon (most probably through the increased affinity of RNA pol. complex to the EM7 promoter of pSEVA2213 plasmid). We inserted the point mutation into a freshly prepared strain EM42 Δ gcd Δ hexR (named PD580) and demonstrated that this strain with pSEVA2213_EM7_gfp plasmid produced 50% more GFP than the same strain without the Ser855Pro mutation in its *rpoD* gene (please, see discussion on pages 29-30 and Supplementary Fig. 11 in the revised manuscript).

Page 29. Line 6. What is the effect of introducing the *rpoD* mutation in the original strain EM42? Do you observe the same increase? Could be tested with the PEM7-GFP construct.

Please, see the previous response.

Page 29. Line 18. "Discomfort" is rather vague. There could be several reasons. Please clarify differences in production between strains. Which are the "reducing cofactor generating enzymes" in PD689 tt L1? Are they NADH or NADPH dependent and what is the redox cost of the pyocyanin pathway?

While the evolved and engineered PD584 L3 strain showed robustness in this last experiment of our study - it grew well and produced the most pyocyanin out of the tested strains - PD689 tt L1 did not outperform either its slowly growing parent strain PD689 or PD310 (Fig. 8 in the revised manuscript). We hypothesize that this low pyocyanin production by PD689 tt L1 is caused by metabolic perturbations (we now use this term instead of more colloquial "discomfort" in the revised manuscript) in this strain. These can stem from confirmed overproduction of the exogenous transporter XylE (overproduction of membrane proteins can disrupt the cell wall - we cite a respective reference: DOI: 10.1016/j.tibtech.2006.06.008) or the implanted *E. coli* xylABE and *tktA*-*tal* pathways and fluxes re-routed due to the *gnd* deletion. The observed upregulation of ROS-reducing enzymes - catalase KatE and glutathione peroxidase - in PD689 tt L1 when compared to PD584 L3 (Fig. 6c in the

revised manuscript and Supplementary File 3) can also reflect the metabolic perturbations of this strain.

Pyocyanin has been studied for its antimicrobial properties. Its mechanisms of action in bacterial cells are complex, among others, it shows strong redox activity (NAD(P)H oxidation, altered redox homeostasis, and ROS generation, DOI: 10.1128/jb.141.1.156-163.1980, DOI: 10.3390/molecules26040927). To the best of our knowledge, the cost of pyocyanin biosynthesis from erythrose 4-phosphate and PEP through the Shikimate pathway is 1 mol NADPH consumed at the step of dihydroxyacetone phosphate conversion to shikimate (please, see the pathway scheme below). Our additional measurements of NAD(H) and NADP(H) levels in PD584 L3 and PD689 tt L1 (please, see the graphs below, raw data are available in the Source data file, sheet Review experiment 3) show that both strains have similar levels of NAD(H) but PD689 tt L1 generates five times more NADP(H). While we can provide an overview of NADPH-producing and consuming reactions in *P. putida* (these are listed in Supplementary File 1, sheet NADPH-NADH Summary), we can only speculate that this big difference in NADP(H) levels indicates a disbalance of redox metabolism in PD689 tt L1 caused by multiple factors mentioned above and that it can affect the ability of the strain to produce pyocyanine and deal with the harsh molecule. Higher NADPH level in PD689 tt L1 could be caused e.g., by the reduced activity of glyoxylate shunt compared to PD584 L3 due to *gnd* deletion (Fig. 6c) and activated TCA cycle reactions including isocitrate dehydrogenase, which is the major NADPH supplying enzyme in *P. putida* KT2440 (DOI: 10.1016/j.ymben.2019.01.008).

However, the effects of pyocyanin in bacterial cells are multiple and a detailed explanation of the above phenomenon would require more experiments and long discussion. As we need to keep our study concise and want to avoid more speculations in the manuscript otherwise added with several new experiments that confirm our key hypotheses, we decided to re-structure and reduce this last part of the Results and discussion section focused on the pyocyanin production in five recombinant *P. putida* strains and leave the reader with a take-home message that accelerated substrate utilization by tailored strains does not necessarily secure enhanced synthesis of a selected bioproduct.

Figure. Concentrations of NADP(H) and NAD(H) measured in selected *P. putida* strains using colorimetric NADP/NADPH Quantitation Kit and NAD/NADH Quantitation Kit, respectively (Merck). Columns represent means \pm standard deviations calculated from two biological replicates (each of two technical replicates).

Figure 7. Visualisation of the pathway would be helpful.

The pyocyanin pathway is quite complex and its individual steps are not yet fully understood. Its visualization in the main text of the manuscript could draw the attention of the reader from the main result, which is the comparison of the bioproduction capacities of five compared *P. putida* strains (Fig. 8 in the revised manuscript). Therefore, we decided to place a simplified schematic illustration of the phenazine biosynthetic pathway (below) in the Supplementary Information as Figure 12.

Figure. Simplified schematic illustration of the phenazine biosynthetic pathway. Abbreviations (metabolites): DHAP, dihydroxyacetone phosphate; E4P, erythrose-4-phosphate; PCA, phenazine-1-carboxylic acid; PEP, phosphoenolpyruvate; (enzymes) PhzC, probable phospho-2-dehydro-3-deoxyheptonate aldolase; PhzG, dihydrophenazinedicarboxylate synthase; PhzF, trans-2,3-dihydro-3-hydroxyanthranilate isomerase; PhzM, phenazine-1-carboxylate N-methyltransferase; PhzS, 5-methylphenazine-1-carboxylate 1-monooxygenase; PhzABDE, phenazine biosynthesis protein.

Reviewers' Comments:

Reviewer #1:

Remarks to the Author:

The revised paper from Dvorak and colleagues addressed all of my concerns. I recommend publication.

The reverse engineering is a really nice addition to the paper – I think that this fully supports and strengthens the story. I appreciate the authors tracking this down.

I still have a minor issue with what has now been reworded to “synthetically-primed adaptation”. Respectfully, I feel like is a bit of a sales pitch, but perhaps this is merely just a personal preference. This is not a major issue, but rather something I felt obligated to point out. My understanding is that the authors are calling SPA the introduction of heterologous genes followed by adaptive laboratory evolution. For what it’s worth, I think that is a fairly standard approach to endow a new function to a microbe through the introduction of heterologous genes then to optimize their function through serial passaging. While there is no need to change this if the authors feel really strongly, assuming I’m understanding what the authors are describing, this does not really merit a new term to me.

Reviewer #2:

Remarks to the Author:

I am satisfied with the revision and recommend publication of the manuscript.

Reviewer #3:

Remarks to the Author:

The authors have adequately addressed points raised in the original review.

Response to minor comment of reviewer #1 on ms. with ID NCOMMS-23-33047A.

Reviewer #1 (Remarks to the Author):

I still have a minor issue with what has now been reworded to “synthetically-primed adaptation”. Respectfully, I feel like is a bit of a sales pitch, but perhaps this is merely just a personal preference. This is not a major issue, but rather something I felt obligated to point out. My understanding is that the authors are calling SPA the introduction of heterologous genes followed by adaptive laboratory evolution. For what it’s worth, I think that is a fairly standard approach to endow a new function to a microbe through the introduction of heterologous genes then to optimize their function through serial passaging. While there is no need to change this if the authors feel really strongly, assuming I’m understanding what the authors are describing, this does not really merit a new term to me.

We thank the reviewer for this comment. Indeed, we strongly believe that this term eloquently and in just three words conveys the important message of this publication, namely that rational interventions in the metabolism of the studied bacterium, including targeted deletions and introduced synthetic constructs, directed/primed the subsequent adaptation to xylose through two parallel evolutionary pathways, which are described in the study for strains PD584 L3 and PD689 tt L1. The correctness of the term “synthetically-primed adaptation” and the whole modified manuscript title “Synthetically-primed adaptation of *Pseudomonas putida* to a non-native substrate D-xylose” was confirmed by a native English speaker and a respected scientist with no ties to the collective of authors. Hence, we wish to keep this title.